# Learning to Act from Actionless Videos through Dense Correspondences

**Po-Chen Ko**[†]
National Taiwan University

**Jiayuan Mao**
MIT CSAIL

**Yilun Du**
MIT CSAIL

**Shao-Hua Sun**
National Taiwan University

**Joshua B. Tenenbaum**
MIT BCS, CBMM, CSAIL

## Abstract

In this work, we present an approach to construct a video-based robot policy capable of reliably executing diverse tasks across different robots and environments from few video demonstrations without using any action annotations. Our method leverages images as a task-agnostic representation, encoding both the state and action information, and text as a general representation for specifying robot goals. By synthesizing videos that "hallucinate" robot executing actions and in combination with dense correspondences between frames, our approach can infer the closed-formed action to execute to an environment without the need of *any* explicit action labels. This unique capability allows us to train the policy solely based on RGB videos and deploy learned policies to various robotic tasks. We demonstrate the efficacy of our approach in learning policies on table-top manipulation and navigation tasks. Additionally, we contribute an open-source framework for efficient video modeling, enabling the training of high-fidelity policy models with four GPUs within a single day.

## 1 Introduction

A goal of robot learning is to construct a policy that can successfully and robustly execute diverse tasks across various robots and environments. A major obstacle is the diversity present in different robotic tasks. The state representation necessary to fold a cloth differs substantially from the one needed for pouring water, picking and placing objects, or navigating, requiring a policy that can process each state representation that arises. Furthermore, the action representation to execute each task varies significantly subject to differences in motor actuation, gripper shape, and task goals, requiring a policy that can correctly deduce an action to execute across different robots and tasks.

One approach to solve this issue is to use images as a task-agnostic method for encoding both the states and the actions to execute. In this setting, policy prediction involves synthesizing a video that depicts the actions a robot should execute (Finn & Levine, 2017; Kurutach et al., 2018; Du et al., 2023), enabling different states and actions to be encoded in a modality-agnostic manner. However, directly predicting an image representation a robot should execute does not explicitly encode the required robot actions to execute. To address this, past works either learn an action-specific video prediction model (Finn & Levine, 2017) or a task-specific inverse-dynamics model to predict actions from videos (Du et al., 2023). Both approaches rely on task-specific action labels which can be expensive to collect in practice, preventing general policy prediction across different robot tasks.

This work presents a method that first synthesizes a video rendering the desired task execution; then, it directly regresses actions from the synthesized video without requiring *any* action labels or task-specific inverse-dynamics model, enabling us to directly formulate policy learning as a video generation problem. Our key insight is that action inference from video in many robotics tasks can be formulated as solving for a rigid 3D transform of objects or points in the generated video. Such a transform can be robustly inferred using off-the-shelf optical flow and segmentation networks, and actions can then be executed from these transforms using off-the-shelf inverse kinematics and motion planners. We illustrate the efficacy of our method across various robotics tasks ranging from table-top assembly, ego-centric object navigation, and real-world robot manipulation in Figure 1.

---

[†]Work done while Po-Chen Ko is a visiting student at MIT. Project page: https://flow-diffusion.github.io/

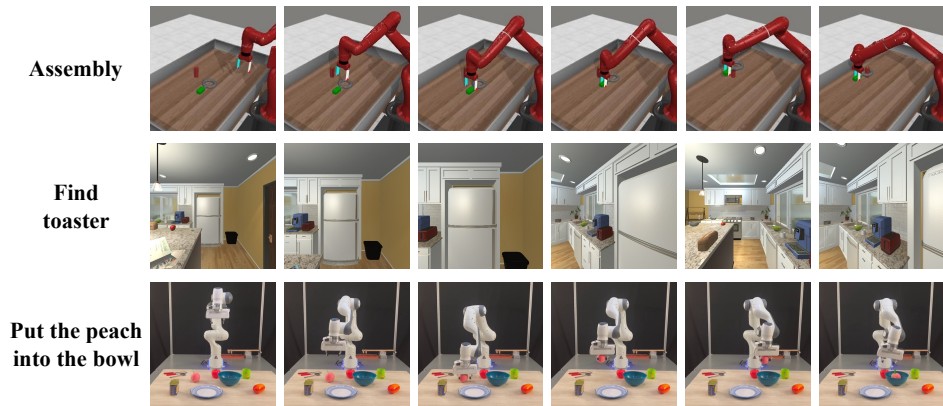

Figure 1: **Diverse Task Execution without Action Labels.** Our approach can execute policies given only synthesized video, without any action labels, across various manipulation, navigation, and real-world tasks.

Another limitation of existing approaches that formulate policy prediction as a video prediction problem is that they suffer from high computational costs during training, requiring the use of over 256 TPU pods (Du et al., 2023), with limited availability of the underlying source code. As a contribution, we provide an open-source codebase for training video policy models. Through a series of architectural optimizations, our framework enables the generation of high-fidelity videos for policy execution, with training accomplished on just 4 GPUs in a single day.

Concretely, this work contributes the following: **(1)** We propose a method to infer actions from video prediction without the need of *any action labels* by leveraging dense correspondences in a video. **(2)** We illustrate how this approach enables us to learn policies that can solve diverse tasks across both table-top manipulation and navigation. **(3)** We present an open-source framework which enables efficient video modeling that enables us to learn policies efficiently on 4 GPUs in a single day.

## 2 RELATED WORK

**Robot Learning from Videos.** A large body of work has explored how to leverage videos for robot learning (Sun et al., 2018; Pari et al., 2022; Nair et al., 2022; Shao et al., 2021; Chen et al., 2021; Bahl et al., 2022; Sharma et al., 2019; Lee & Ryoo, 2017; Du et al., 2023; Chethan et al., 2023; Karamcheti et al., 2023). One approach relies upon using existing video datasets to construct effective visual representations (Pari et al., 2022; Nair et al., 2022; Karamcheti et al., 2023). Alternatively, goal or subtask information for robotic execution may be extracted for videos (Shao et al., 2021; Chen et al., 2021; Chethan et al., 2023; Bahl et al., 2022; Sharma et al., 2019; Lee & Ryoo, 2017; Sivakumar et al., 2022) or used as a dynamics model for planning (Finn & Levine, 2017; Kurutach et al., 2018). The absence of rewards and action labels distinguishes our work from offline RL (Levine et al., 2020). Most similar to our work, in UniPi (Du et al., 2023), policy prediction may directly be formulated as a text-conditioned video generation problem. Our approach extends UniPi and illustrates how dense correspondences enable action inference without any explicit action labels. Another work with a similar high-level idea to ours (Bharadhwaj et al., 2023) predicts hand poses from videos and uses them directly for control, while we infer actions from object-centric trajectories. While hand poses contain more details of manipulator-object interactions, object-centric actions may help cross-embodiment transfer.

**Leveraging Dense Correspondences.** Dense correspondences have emerged as an effective implicit parameterization of actions and poses (Florence et al., 2018; Manuelli et al., 2022; Yen-Chen et al., 2022; Simeonov et al., 2022; 2023; Chun et al., 2023; Sundaresan et al., 2020; Ryu et al., 2023). Given dense correspondences in 2D (Florence et al., 2018; Manuelli et al., 2022; Sundaresan et al., 2020; Yen-Chen et al., 2022) of 3D (Simeonov et al., 2022; 2023; Chun et al., 2023; Ryu et al., 2023) both object and manipulator poses may be inferred by solving for rigid transforms given correspondences. Our approach uses dense correspondences between adjacent frames of synthesized videos to calculate object of scene transformations and then infer robot actions.

**Learning from Observation.** In contrast to imitation learning (learning from demonstration: Osa et al., 2018; Kipf et al., 2019; Ding et al., 2019; Fang et al., 2019; Mao et al., 2022; Wang et al., 2023), which assumes access to expert actions, learning from observation methods (Torabi et al., 2019b; 2018; 2019a; Lee et al., 2021; Karnan et al., 2022) learn from expert state sequences (*e.g.*,

Figure 2: **Overall framework of AVDC.** (a) Our model takes the RGBD observation of the current environmental state and a textual goal description as its input. (b) It first synthesizes a video of *imagined* execution of the task using a diffusion model. (c) Next, it estimates the optical flow between adjacent frames in the video. (d) Finally, it leverages the optical flow as dense correspondences between frames and the depth of the first frame to compute $SE(3)$ transformations of the target object, and subsequently, robot arm commands.

video frames). Action-free pre-training methods (Baker et al., 2022; Escontrela et al., 2023) extract knowledge from unlabeled videos and learn target tasks through RL. extcolorblackFor example, a recent approach involves learning value functions by pre-training on existing video datasets (Chethan et al., 2023). Despite encouraging results, these methods require interacting with environments, which may be expensive or even impossible. In contrast, our proposed method does not require environmental interactions and therefore is more applicable.

# 3 ACTIONS FROM VIDEO DENSE CORRESPONDENCES

The architecture of our proposed framework, Actions from Video Dense Correspondences (AVDC), is depicted in Figure 2. AVDC consists of three modules. Given the initial observation (*i.e.*, an RGBD image of the scene and a textual task description), we first employ a video synthesis model to generate a video that implicitly captures the sequence of required actions (Section 3.1). Then, we use a flow prediction model to estimate the optical flow of the scene and objects from the synthesized video (Section 3.2). Finally, leveraging the initial depth map and predicted optical flows, we reconstruct the movements of objects for manipulation or robots for navigation, described in Section 3.3.

## 3.1 TEXT-CONDITIONED VIDEO GENERATION

Our text-conditioned video generation model is a conditional diffusion model. The diffusion model takes the initial frame and a text description as its condition and learns to model the distribution of possible future frames. Throughout this paper, our video generation model predicts a fixed number of future frames ($T = 8$ in our experiments).

The diffusion model aims to approximate the distribution $p(img_{1:T}|img_0, txt)$, where $img_{1:T}$ represents the video frames from time step 1 to $T$, $img_0$ denotes the initial frame, and *txt* represents the task description. We train a denoising function $\epsilon_\theta$ that predicts the noise applied to $img_{1:T}$ given the perturbed frames. Given the Gaussian noise scheduling $\beta_t$, our overall objective is,

$$\mathcal{L}_{\text{MSE}} = \left\| \epsilon - \epsilon_\theta \left( \sqrt{1 - \beta_t} img_{1:T} + \sqrt{\beta_t}\epsilon, t \mid txt \right) \right\|^2,$$

where $\epsilon$ is sampled from a multivariate standard Gaussian distribution, and $t$ is a randomly sampled diffusion step $t$. A main practical challenge with training such video diffusion models is that they are usually computationally expensive. For example, the closest work to us, UniPi (Du et al. (2023)), requires over 256 TPU pods to train. In this paper, we build a high-fidelity video generation model that can be trained on 4 GPUs in a single day through a series of architectural optimizations. Section G presents complexity analyses and how the process can be significantly accelerated.

Our model is a modified version of the image diffusion model proposed by Dhariwal & Nichol (2021), built upon U-Net (Ronneberger et al., 2015), as illustrated in Figure 3a. The U-Net consists of the same number of downsample blocks and upsample blocks. To enhance consistency with the initial frame, we concatenate the input condition frame $img_0$ to all future frames $img_{1:T}$. To encode the text, we use a CLIP-Text (Radford et al., 2021) encoder to obtain a vector embedding and combine it into the video generative model as additional inputs to individual downsampling and upsampling blocks.

Importantly, we use a factorized spatial-temporal convolution similar to the model from Ho et al. (2022), within each ResNet block (He et al., 2016). As shown in Figure 3b, in our approach, the 5D input feature map with shape $(B, H, W, T, C)$, where $B$ is the batch size, $H$ and $W$ represent the spatial dimensions, $T$ is the number of time frames, and $C$ denotes the number of channels,

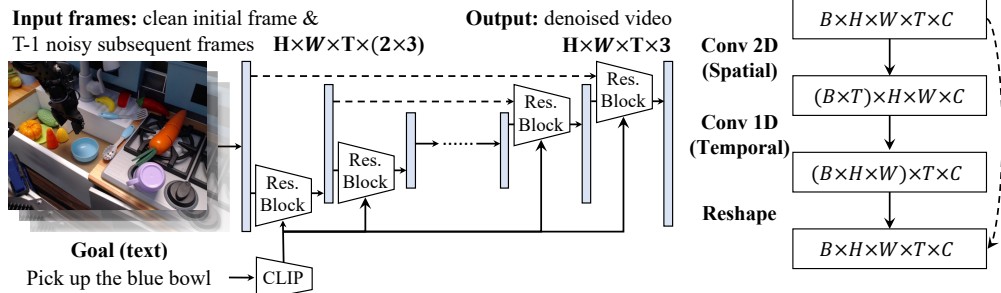

(a) U-Net architecture for our base diffusion model        (b) Factorized spatial-temporal ResNet block

Figure 3: **Network architecture of our video diffusion model.** (a) We use an U-Net architecture following Dhariwal & Nichol (2021) but extending it to videos. (b) We use a factorized spatial-temporal convolution kernel Sun et al. (2015) as the basic building block. Dashed lines in both figures represent residual connections (He et al., 2016).

undergoes two consecutive convolution operations. First, we apply a spatial convolution identically and independently to each time step $t = 1, 2, \cdots, T$. Then, we employ a temporal convolution layer identically and independently at each spatial location. This factorized spatial-temporal convolution replaces conventional 3D convolution methods, leading to significant improvements in training and inference efficiency without sacrificing generation quality. More details on the model architecture and training can be found in Section F.

## 3.2 FLOW PREDICTION

To regress actions from predicted videos, we leverage flow prediction as an intermediate representation. We employ off-the-shelf GMFlow, a transformer architecture specifically designed for optical flow prediction (Xu et al., 2022). Given two consecutive frames $img_i$ and $img_{i+1}$ predicted by the video diffusion model, GMFlow predicts the optical flow between two images as a vector field on the image, which is essentially a pixel-level *dense correspondence map* between two frames. This allows us to track the movement of each input pixel with a simple integration of this vector field over time.

Alternatively, one could train diffusion models to directly predict the flow by first preprocessing training videos with the flow prediction model. However, in our experiments, we encountered challenges in optimizing such models and observed that they failed to match the performance achieved by the two-stage inference pipeline. We conjecture that this difficulty arises from the lack of spatial and temporal smoothness in flow fields. For instance, the flow field is sparse when only a single object moves. Consequently, the Gaussian diffusion model may not be the optimal model for flow distributions. We empirically compare two alternatives in subsequent experiments.

## 3.3 ACTION REGRESSION FROM FLOWS AND DEPTHS

Based on the predicted flow, which essentially gives us a dense prediction of pixel movements, we can reconstruct object movements and robot movements in the video. Our key insight is to, given the 3D information (depth) of the input frame and dense pixel tracking, reconstruct a sequence of 3D rigid transformations for each object. In this work, we explore two different settings: predicting object transformations assuming a fixed camera (fixed-camera object manipulation) and predicting camera (robot) movement assuming a static scene (visual navigation).

**Predict object-centric motion**. We first consider predicting 3D object motions in videos assuming a fixed camera. We represent each object as a set of 3D points $\{x_i\}$. The points corresponding to the object of interest will be extracted by external segmentation methods, such as a pretrained image segmentation model, or simply specified by the human. Given the camera intrinsic matrix and the input RGBD image, we can compute the initial 3D positions of these points. Let $T_t$ denote the rigid body transformation of the object at time step $t$ relative to the initial frame. We can express the projection of a 3D point onto the image plane at time step $t$ as $KT_t x = (u_t, v_t, d_t)$, where $K$ is the camera intrinsic matrix. Furthermore, the projected 2D point on frame $t$ is thus $(u_t/d_t, v_t/d_t)$.

The optical flow tracking provides us with the projection of the same point in frame $t$, specifically $u_t/d_t$ and $v_t/d_t$. By tracking all points in $\{x_i\}$, we can find the optimal transformation $T_t$ that minimizes the following L2 loss:

$$\mathcal{L}_{\text{Trans}} = \sum_i \left\| u_t^i - \frac{(KT_t x_i)_1}{(KT_t x_i)_3} \right\|_2^2 + \left\| v_t^i - \frac{(KT_t x_i)_2}{(KT_t x_i)_3} \right\|_2^2,$$

where $(u_t^i, v_t^i)$ is the corresponding pixel of point $x_i$ in frame $t$, and $(KT_t x_i)_i$ denotes the $i$-th entry of the vector. It's worth noting that even if we do not directly observe $d_t$, this loss function remains well-formed based on the assumption that $T_t$ represents a rigid body transformation.

During execution, we first extract the mask of the object to manipulate and use the dense correspondences in predicted videos to compute the sequence of rigid body transformations for the object. Next, given inferred object transformations, we can use existing off-the-shelf robotics primitives to generalizably infer actions in the environment. In particular, if the object is graspable, we randomly sample a grasp on the object and then compute the target robot end-effector pose based on the target object pose and the grasping pose. When the object is not directly graspable (*e.g.*, a door), we similarly sample a contact point and use a push action to achieve the target object transformation.

We treat the grasp/contact point as the first subgoal. Then, we iteratively apply the computed transformation on the current subgoal to compute the next subgoal until all subgoals are computed. Next, we use a position controller to control the robot to reach the subgoals one by one. More details on inferring robot manipulation actions can be found in Section H.1. In contrast to our approach, directly learning explicitly regress actions using a learned inverse dynamics requires a substantial number of action labels so that a neural network can learn existing knowledge such as inverse dynamics, grasping and motion-planning.

**Inferring Robot Motion.** The similar algorithm can also be applied to predict robot (*i.e.*, the camera) motion assuming all objects are static. Due to the duality of camera motion and object motion, we can use exactly the same optimization algorithm to find $T_t$ (object-centric motion), and the camera motion $C_t = (T_t)^{-1}$. Concretely, we make the following modifications to adapt AVDC to navigation tasks. (1) The video diffusion model is trained to duplicate the last frame once the object is found. (2) Instead of tracking objects, we utilize the optical flow of the whole frame to estimate the rigid transformations between frames. (3) Based on the calculated rigid transformations, we simply map the transformations to the closest actions, detailed in Section H.2.

**Depth Estimation.** We can reconstruct 3D object or robot trajectories solely from the depth map of the initial frame (*i.e.*, the subsequent depth maps is not required). By leveraging dense correspondences between frames and assuming rigid object motion, we can reconstruct accurate 3D trajectories. This holds significant advantages as it enables us to train video prediction models exclusively using RGB videos, allowing for learning from online sources like YouTube, and only requires an RGB-D camera (or monocular depth estimator) at execution time. By eliminating the dependence on depth maps from subsequent frames, our system is significantly more adaptable to various data sources.

**Replanning Strategy.** After inferring the object or robot trajectories, we can execute the trajectory using a position controller in an open-loop manner. Yet, it can suffer from accumulated errors. As the planning horizon increases, the accuracy of predicted object locations diminishes due to combined errors in video synthesis and flow prediction. To mitigate this issue, we propose a replanning strategy. If the robot movement is smaller than 1mm over 15 consecutive time steps while the task has not been fulfilled, we re-run our video generation and action prediction pipeline from the current observation.

## 4 EXPERIMENTS

We describe the baselines and the variants of our proposed method AVDC in Section 4.1. Then, we compare AVDC to its variants and the baselines on simulated robot arm manipulation tasks in Meta-World (Figure 4a) in Section 4.2 and simulated navigation tasks in iTHOR (Figure 4b) in Section 4.3. Note that although it is possible to obtain ground-truth actions from demonstrations in these two domains, our method does not use these actions; instead, these actions are only used by the baselines to provide an understanding of the task difficulty. Then, Section 4.4 evaluate the ability of AVDC to control robots by learning from out-of-domain human videos without actions, as illustrated in Figure 4c. In Section 4.5, we leverage the Bridge dataset (Figure 4d) and evaluate AVDC on real-world manipulation tasks with a Franka Emika Panda robot arm (Figure 4e). Extended qualitative results can be found in Section B and additional experimental details can be found in Section H.

### 4.1 BASELINES AND VARIANTS OF AVDC

**Baselines.** We compare AVDC to a multi-task behavioral cloning (BC) baseline given access to a set of expert actions from all videos ($15,216$ labeled frame-action pairs in Meta-World and $5,757$ in iTHOR), which are unavailable to our method. This baseline encodes the RGB observation to a

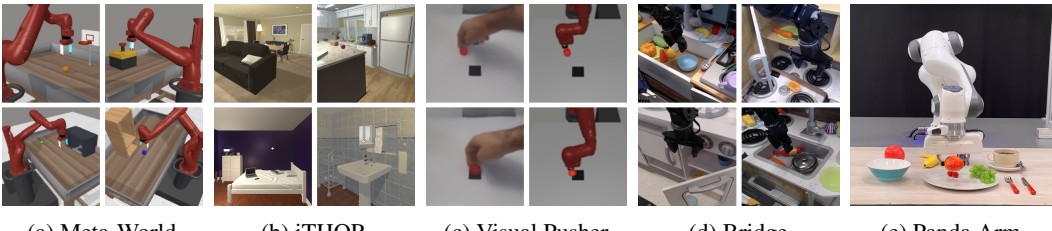

|              |              |              |              |              |
|--------------|--------------|--------------|--------------|--------------|
| (a) Meta-World | (b) iTHOR   | (c) Visual Pusher | (d) Bridge | (e) Panda Arm |

Figure 4: **Environments & Tasks. (a) Meta-World** is a simulated benchmark featuring various tasks with a Sawyer robot arm. **(b) iTHOR** is a simulated benchmark for embodied common sense reasoning. We adopt its object navigation task, requiring navigating to target objects located in different rooms. **(c) Visual Pusher** is a real-world video dataset with 195 human pushing videos. **(d) Bridge** is a real-world video dataset comprised of 33, 078 robot demonstrations conducting various kitchen tasks. **(e) Panda Arm** is a real-world pick-and-place tabletop environment with a Franka Emika Panda robot arm.

feature vector with a ResNet-18 (He et al., 2016). Then, the feature vector is concatenated with a one-hot encoded camera ID and a task representation encoded by the CLIP-Text model (Radford et al., 2021). The concatenated representation is then fed to a 3-layer MLP, which produces an action. We explore initializing the weights of ResNet-18 from scratch (BC-Scratch) or from the pre-trained parameters of R3M (Nair et al., 2022) (BC-R3M).

Additionaly, we experimented with Diffusion Policy (Chi et al., 2023), which also leverages denoising diffusion, but directly predicts actions instead of video frames like we did. We followed the setting used by most of the experiments in the original paper. More details are described in Section H.1.4.

We also implement UniPi (Du et al., 2023), a learning-from-video method that learns an inverse dynamics model to generate actions from videos, as a baseline. Specifically, UniPi infers actions from the videos synthesized by AVDC. Since the exact number of steps between two generated frames in our model may vary across different episodes, we modify the inverse dynamics model to output an additional binary label indicating whether to switch to the next frame of synthesized video plans. This predictor can be trained with the demonstrations (with actions) used to train the BC baselines.

**AVDC and its Variants.** We compare AVDC to its variants that also predict dense correspondence.

- **AVDC (Flow)** learns to directly predict the optical flow between frames as described in Section 3.2. We include this variant to justify our 2-stage design, which synthesizes a video and then infers optical flows between each pair of frames.
- **AVDC (No Replan)** is the opened-loop variant of our proposed method, which synthesizes a video, infers flows, produces a plan, executes it, and finishes, regardless of if it fails or succeeds. We include this variant to investigate whether our replanning strategy is effective.
- **AVDC (Full)** is our proposed method in full, employing the 2-stage design and can replan.

**Additional Ablation Studies and Experiments.** We also include additional ablation studies on the effect of first-frame conditioning in video generation and different text encoders (*e.g.*, CLIP and T5) in Section E, a study of extracting object mask with existing segmentation model in Section D.1, an experiment training BC with more data in Section D.2, using object masks extensively as proxy for actions in Section D.3, and a quantitative quality analysis on the synthesized videos in Section D.4.

## 4.2 META-WORLD

**Setup.** Meta-World (Yu et al., 2019) is a simulated benchmark featuring various manipulation tasks with a Sawyer robot arm. We include 11 tasks, and for each task, we render videos from 3 different camera poses. The same set of camera poses is used for training and testing. We collect 5 demonstrations per task per camera position, resulting in total 165 videos. To isolate the problem of learning object manipulation skills, for our methods and all its variants, we provide the ground-truth segmentation mask for the target object. We include an additional study on using external segmentation models in Appendix D.1.

Each policy is evaluated on each task with 3 camera poses, each with 25 trials. A policy succeeds if it reaches the goal state within the maximum environment step and fails otherwise. The positions of the robot arm and objects are randomized when each episode begins. The result is reported in Table 1.

**Comparison to Baselines.** Our method AVDC (Full) consistently outperforms the two BC baselines (BC-Scratch and BC-R3M) and UniPi on all the tasks by a large margin. Furthermore, AVDC

|  | door-open | door-close | basketball | shelf-place | btn-press | btn-press-top |
|---|---|---|---|---|---|---|
| BC-Scratch | 21.3% | 36.0% | 0.0% | 0.0% | 34.7% | 12.0% |
| BC-R3M | 1.3% | 58.7% | 0.0% | 0.0% | 36.0% | 4.0% |
| UniPi (With Replan) | 0.0% | 36.0% | 0.0% | 0.0% | 6.7% | 0.0% |
| Diffusion Policy | 45.3 % | 45.3 % | 8.0 % | 0.0 % | 40.0 % | 18.7 % |
| AVDC (ID) | 0.0% | 36.0% | 0.0% | 0.0% | 0.0% | 0.0% |
| AVDC (Flow) | 0.0% | 0.0% | 0.0% | 0.0% | 1.3% | **40.0**% |
| AVDC (No Replan) | 30.7% | 28.0% | 21.3% | 8.0% | 34.7% | 17.3% |
| AVDC (Full) | **72.0**% | **89.3**% | **37.3**% | **18.7**% | **60.0**% | 24.0% |
|  | faucet-close | faucet-open | handle-press | hammer | assembly | **Overall** |
| BC-Scratch | 18.7% | 17.3% | 37.3% | 0.0% | 1.3% | 16.2% |
| BC-R3M | 18.7% | 22.7% | 28.0% | 0.0% | 0.0% | 15.4% |
| UniPi (With Replan) | 4.0% | 9.3% | 13.3% | 4.0% | 0.0% | 6.1% |
| Diffusion Policy | 22.7% | **58.7**% | 21.3% | 4.0% | 1.3% | 24.1% |
| AVDC (ID) | 4.0% | 9.3% | 13.3% | 4.0% | 0.0% | 6.1% |
| AVDC (Flow) | 42.7% | 0.0% | 66.7% | 0.0% | 0.0% | 13.7% |
| AVDC (No Replan) | 12.0% | 17.3% | 41.3% | 0.0% | 5.3% | 19.6% |
| AVDC (Full) | **53.3**% | 24.0% | **81.3**% | **8.0**% | **6.7**% | **43.1**% |

Table 1: **Meta-World Result.** We report the mean success rate across tasks. Each entry of the table shows the average success rate aggregated from 3 camera poses with 25 seeds for each camera pose.

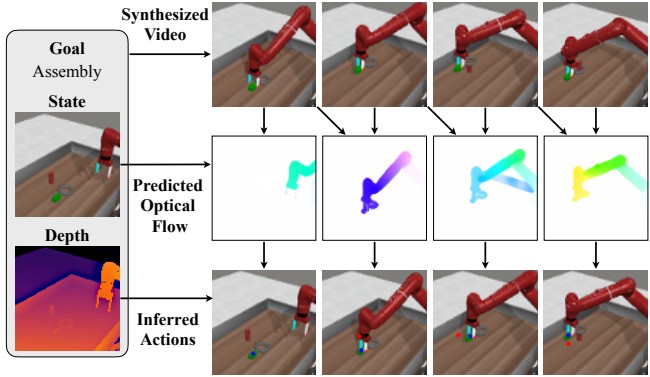

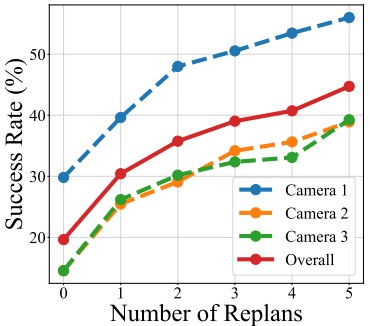

Figure 5: **Qualitative Results on Meta-World.** AVDC can reliably synthesize a video, predict optical flow between frames, and infer and execute actions to fulfill the assembly task. Current subgoals (•) and next subgoals (•) are rendered in inferred action visualizations.

Figure 6: **Number of Replanning Steps vs. Success Rate.** Our method AVDC achieves higher success rates across all viewing angles with more replanning trials, justifying the effectiveness of our replanning strategy.

(Full) also outperforms the Diffusion Policy in 10 out of 11 tasks and in overall performance by a significant margin. This indicates that the tasks are still very challenging, even with access to expert actions. Note that AVDC (Full) is able to solve the task "hammer," which involves using tools, with performance surpassing all baselines. This is done by predicting actions based on tool motions.

**Comparing AVDC Variants.** AVDC (Flow) performs the best on button-press-topdown and achieves reasonable performance on faucet-close and handle-press, while performing very poorly on the rest of the tasks. As described in Section 3.2, the diffusion model employed in this work may not be optimal for flow prediction. Also, AVDC (Full) consistently outperforms AVDC (No Replan), justifying the effectiveness of our closed-loop design, enabling replanning when the policy fails.

**Intermediate Outputs.** To provide insights into the pipeline of AVDC, we visualized the synthesized video, predicted optical flow, and inferred actions (*i.e.*, motion planning) in Figure 5. Our diffusion model synthesizes a reasonable video showing the robot arm picking up the nut and placing it onto the peg. The optical flow predicted from video frames accurately captures the robot arm's motions. Then, based on the predicted flow, the inferred actions can reliably guide the arm to fulfill the task.

**Effect of Replanning Trials.** We investigate how varying the maximum number of replanning step affects the performance of AVDC. As presented in Figure 6, the success rate consistently increases with more replanning trials, demonstrating the effectiveness of our proposed replanning strategy.

**Failure Modes.** The primary failure mode we observed is the errors made by the optical flow tracking model, partially because these models are not trained on any in-domain data. Since the prediction resolution is not very high in our experiments, small pixel-level errors in tracking small objects would

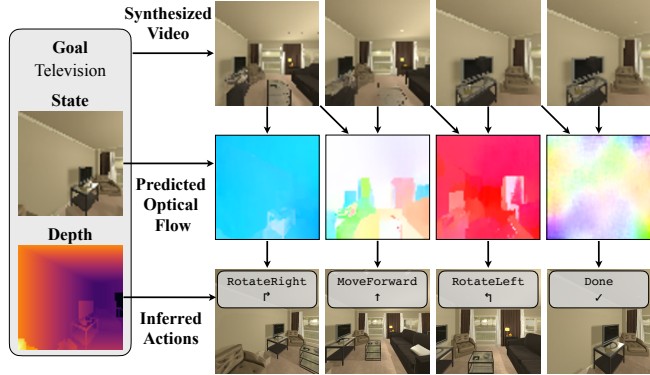

| Room | BC-Scratch | BC-R3M | AVDC |
|---|---|---|---|
| Kitchen | 1.7% | 0.0% | **26.7**% |
| Living Room | 3.3% | 0.0% | **23.3**% |
| Bedroom | 1.7% | 1.7% | **38.3**% |
| Bathroom | 1.7% | 0.0% | **36.7**% |
| Overall | 2.1% | 0.4% | **31.3**% |

Table 2: **iTHOR Result.** We report the mean success rate, aggregated from 3 types of objects per room with 20 episodes per object. Both the two BC baselines fail to achieve meaningful performance on the iTHOR object navigation tasks. On the other hand, AVDC performs reasonably with a 31.3% average success rate.

Figure 7: **Qualitative Results on iTHOR.** AVDC can reliably synthesize a video, predict optical flow between frames, and infer and execute actions to navigate to the television.

result in large errors in the 3D space. We believe that by directly increasing the resolution of video synthesis or by training an in-domain optical flow model, we can improve the performance.

### 4.3 iTHOR

**Setup.** iTHOR (Kolve et al., 2017) is a simulated benchmark for embodied common sense reasoning. We consider the object navigation tasks for evaluation, where an agent randomly initialized into a scene learns to navigate to an object of a given type (*e.g.*, toaster, television). At each time step, the agent observes a 2D scene and takes one of the four actions: MoveForward, RotateLeft, RotateRight, and Done. We chose 12 different objects to be placed at 4 type of rooms (*e.g.*, kitchen, living room). No object segmentation is required in this navigation task.

Each policy is evaluated on 12 object navigation tasks distributed in 4 different types of rooms (3 tasks for each room). A policy succeeds if the target object is in the agent's sight and within a 1.5m distance within the maximum environment step or when Done is predicted and fails otherwise. The position of the agent is randomized at the beginning of each episode. The result is reported in Table 2.

**Comparison to Baselines.** Our proposed method AVDC can find target objects in different types of rooms fairly often (31.3%), while the two BC baselines fail entirely. BC-R3M with a pre-trained ResNet-18 performs worse than BC-Scratch, which can be attributed to the fact that R3M is pre-trained on robot manipulation tasks and might not be suitable for visual navigation tasks.

**Intermediate Outputs.** The intermediate outputs produced by AVDC are presented in Figure 7. The diffusion model can synthesize video showing an agent navigating to the target object. Then, desired agent movements can be easily inferred from the predicted optical flow, resulting in the ease of mapping the flow to MoveForward, RotateLeft, or RotateRight. When no flow is predicted, it indicates the agent has found the object and selects Done as the predicted action.

### 4.4 CROSS-EMBODIMENT LEARNING: FROM HUMAN VIDEOS TO ROBOT EXECUTION

We aim to examine if AVDC can achieve cross-embodiment learning, *e.g.*, leverage *human* demonstration videos to control *robots* to solve tasks.

**Setup.** We evaluate our method with Visual Pusher tasks (Schmeckpeper et al., 2021; Zakka et al., 2022). Specifically, we learn a video diffusion model from only actionless human pushing data (198 videos), with the same U-net architecture used in Meta-World experiments and trained the model for 10k steps. Then, we evaluate AVDC on simulated robot pushing tasks *without any fine-tuning*.

**Results.** AVDC exhibits strong zero-shot transfer capability, achieving a 90% zero-shot success rate out of 40 runs. This indicates that AVDC can perform cross-embodiment learning — utilizing out-of-domain human videos to achieve reliable robot execution. A synthesized video and the corresponding robot execution are illustrated in Figure 8.

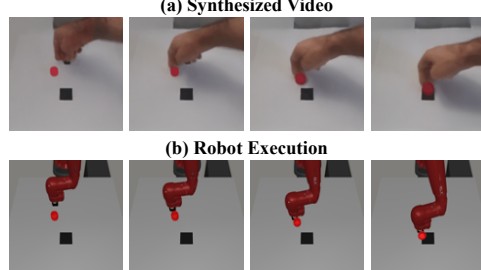

Figure 8: **Qualitative Results on Visual Pusher.** AVDC can **(a) synthesize video plans** by watching *human* demonstrations and **(b) infer actions** to control the *robot* without any fine-tuning.

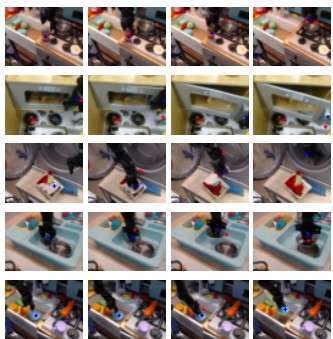
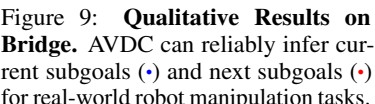

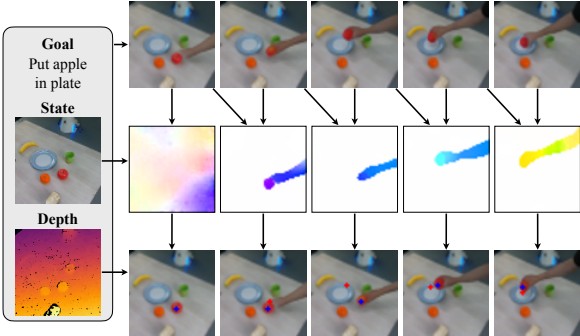

Figure 9: **Qualitative Results on Bridge.** AVDC can reliably infer current subgoals (•) and next subgoals (•) for real-world robot manipulation tasks.

Figure 10: **Qualitative Results on Franka Emika Panda.** AVDC can reliably synthesize a video, predict optical flow between frames, and infer and execute actions to fulfill the assembly task. Current subgoals (•) and next subgoals (•) are rendered in the bottom row.

### 4.5 REAL-WORLD FRANKA EMIKA PANDA ARM WITH BRIDGE DATASET

We aim to investigate if our proposed framework AVDC can tackle real-world robotics tasks. To this end, we train our video generation model on the Bridge dataset (Ebert et al., 2022), and perform evaluation on a real-world Franka Emika Panda tabletop manipulation environment.

**Setup.** The Bridge dataset (Ebert et al., 2022) provides $33,078$ teleoperated WidowX 250 robot demonstrations of various kitchen tasks captured by a web camera without depth information. Our real-world setup comprises a Franka Emika Panda robot arm and an Intel Realsense D435 RGBD camera mounted at a fixed frame relative to the table. Due to the differences in camera FOVs and the environmental setup, directly applying the video generative model trained on Bridge to our setup does not generalize well. We thus fine-tuned the diffusion model with 20 human demonstrations collected with our setup. In our real-world evaluation, we assume that the target object can be grasped using a top-grasp so that no reorientation of the target object is needed. Note that both the Bridge dataset and our human demonstration datasets do not contain any action label relevant to our robot: Bridge is based on a different robot model and our tabletop videos are human hand manipulation videos.

**Zero-Shot Generalization of Bridge Model.** We found that the video diffusion model trained on Bridge videos can reasonably generalize to real scenes without fine-tuning, as discussed in Section C.

**Results.** The predicted object motion qualitative results on the Bridge dataset are presented in Figure 9. AVDC can reliably synthesize videos, predict optical flow, identify target objects, and infer actions. Figure 10 presents the visualizations of planned robot trajectories, showcasing the successful deployment of our system. More qualitative results can be found in Section B. We also quantitatively evaluated the entire pipeline. To this end, we set up 10 scenes with different initial object configurations and tasks. Each task requires a pick-and-place of an object of a specified category (*e.g.*, apple) to a container (*e.g.*, plate). The results are detailed in Section H.3.

## 5 DISCUSSION

**Limitations.** The proposed AVDC, while being successful in diverse simulated and real-world settings, faces several challenges. First, the algorithm may lose track of objects heavily occluded by the robot arm or struggle with optical flow prediction when there are rapid lighting changes or significant object movements. Additionally, our current implementation is not adept at handling tasks with deformable objects, requiring future work to develop new strategies for tracking or representing these objects, such as key-point-based tracking. Real-world manipulation tasks, which often require predicting grasps or contact points, are also challenging due to the disparity between human hands and different robot hands, necessitating the integration of specialized manipulation algorithms such as grasp prediction modules (Sundermeyer et al., 2021). Lastly, force information in RGB videos is unobtainable. Future work may consider leveraging real-world interaction data to address this.

**Conclusion.** This work presents an approach to learning to act directly in environments given only RGB video demonstrations by exploiting dense correspondence between synthesized video frames. We illustrate the general applicability of our approach in both simulated and real-world manipulation and navigation tasks and cross-embodiment settings. We further present an open-source implementation for fast and efficient video modeling. We hope our work inspires further work on learning from videos, which can be readily found on the internet and readily captured across robots.

**Acknowledgement.**. We thank anonymous reviewers for their valuable comments. We gratefully acknowledge support from ONR MURI grant N00014-16-1-2007; from the Center for Brain, Minds, and Machines (CBMM, funded by NSF STC award CCF-1231216); from NSF grant 2214177; from Air Force Office of Scientific Research (AFOSR) grant FA9550-22-1-0249; from ONR MURI grant N00014-22-1-2740; from ARO grant W911NF-23-1-0034; from the MIT-IBM Watson AI Lab; from the MIT Quest for Intelligence; and from the Boston Dynamics Artificial Intelligence Institute. This project was partially supported by the National Science and Technology Council in Taiwan (NSTC 111-2221-E-002-189). Shao-Hua Sun was partially supported by the Yushan Fellow Program by the Ministry of Education, Taiwan. Any opinions, findings, and conclusions or recommendations expressed in this material are those of the authors and do not necessarily reflect the views of our sponsors.

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

APPENDIX

# Table of Contents

## A CODE

The code for reproducing our results is included in `./codebase_AVDC` directory of the attached supplementary .zip file.

## B EXTENDED QUALITATIVE RESULTS

Our supplementary website presents additional qualitative results, including

- **Synthesized Videos**: Meta-World, iTHOR, Visual Pusher, and Bridge.
- **Task Execution Videos**: Meta-World, iTHOR, Visual Pusher, and the real-world Franka Emika Panda tasks.

## C ZERO-SHOT GENERALIZATION ON REAL-WORLD SCENES

While most tasks in the Bridge data were recorded in toy kitchens, we found that the video diffusion model trained on this dataset already can generalize to complex real-world kitchen scenarios, producing reasonable videos given RGB images and textual task descriptions. Examples of the synthesized videos can be found on our supplementary website.

**Successful mask prediction**

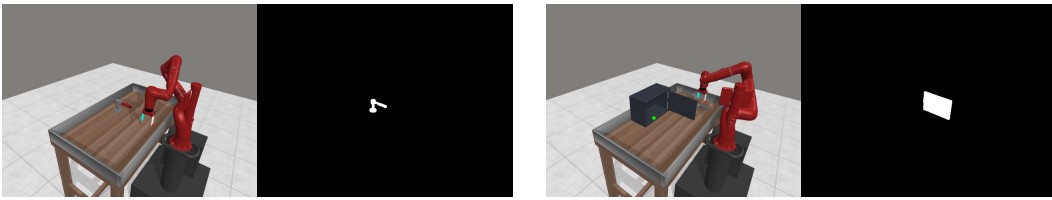

Input text: small faucet handle                    Input text: gray door

**Failed mask prediction**

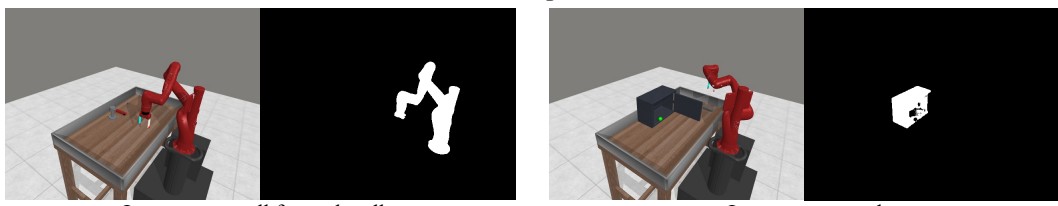

Input text: small faucet handle                    Input text: gray door

Figure 11: **Object Mask with Segmentation models.** Successful and failed object masks extracted by Language Segment Anything.

## D  EXTENDED EXPERIMENTS

### D.1  OBJECT MASK WITH SEGMENTATION MODELS

We experimented with utilizing existing segmentation methods to extract the object mask for our action regression algorithm. We employed Language Segment-Anything (Medeiros, 2023), which is based on GroundingDINO (Liu et al., 2023) and Segment-Anything (Kirillov et al., 2023). The experiment was conducted in our Meta-World setting, using the predicted object mask instead of the GT object mask. In this setup, the achieved success rate averaged over all 11 tasks is 34.5%, which is 8.6% lower than the success rate using the GT object mask (43.1%). The performance drop is attributed to incorrect object masks produced by the object segmentation model. Qualitative object segmentation results are presented in Figure 11.

### D.2  BC WITH MORE DATA

We experimented with training plain BC with more data. The results are reported below.

| | Door-Open | Door-Close | Basketball | Shelf-Place | Btn-Press | Btn-Press-Top |
|---|---|---|---|---|---|---|
| AVDC-165 | 72.0% | 89.3% | 37.3% | 18.7% | 60.0% | 24.0% |
| BC-165 | 21.3% | 36.0% | 0.0% | 0.0% | 34.7% | 12.0% |
| BC-330 | 21.3% | 65.3% | 0.0% | 0.0% | 45.3% | 21.3% |
| BC-660 | 61.3% | 72.0% | 0.0% | 1.3% | 77.3% | 49.3% |
| BC-1650 | 96.0% | 81.3% | 0.0% | 0.0% | 96.0% | 85.3% |
| | Faucet-Close | Faucet-Open | Handle-Press | Hammer | Assembly | Overall |
| AVDC-165 | 53.3% | 24.0% | 81.3% | 8.0% | 6.7% | 43.1% |
| BC-165 | 18.7% | 17.3% | 37.3% | 0.0% | 1.3% | 16.2% |
| BC-330 | 44.0% | 29.3% | 29.3% | 2.7% | 0.0% | 23.5% |
| BC-660 | 77.3% | 77.3% | 62.7% | 10.7% | 0.0% | 44.5% |
| BC-1650 | 93.3% | 94.7% | 86.7% | 12.0% | 0.0% | 58.7% |

Table 3: BC with More Data

Table 3 shows that plain BC needs 20 videos per view (in total 660 videos with action labels, compared to AVDC trained with 165 videos without action labesls) to perform similarly to our method AVDC (43.1% overall). That said, BC needs around 4 times more videos and action labels than our method,

which highlights the efficiency of our proposed method. More importantly, BC still cannot learn most tasks that require grasps (e.g., pick and place, use tools) even with 50 demonstrations per view (BC-1650).

It's important to emphasize that we included the performance of BC to calibrate the difficulty of the tasks, and BC has access to action labels that are not accessible to our proposed method; therefore, this is not a fair comparison.

### D.3 OBJECT MASK AS PROXY FOR ACTION

AVDC have access to object masks only at test time. However, it's also possible to obtain object mask during training time by running the existing object segmentation model on the training dataset. Such extensive use of object mask can serve as a proxy for actions. To provide an idea of the performance of using object masks as a proxy for actions, we have conducted experiments with the following setting: We trained a model that takes in a segmented object and directly predicts optical flow within the segmentation (without diffusion). Then, we used the same procedure as AVDC to calculate actions. The results are presented as follows.

|  | Door-Open | Door-Close | Basketball | Shelf-Place | Btn-Press | Btn-Press-Top |
|---|---|---|---|---|---|---|
| Object Mask Proxy | 1.3% | 20.0% | 0.0% | 0.0% | 12.0% | 2.7% |
| AVDC (Full) | 72.0% | 89.3% | 37.3% | 18.7% | 60.0% | 24.0% |
|  | Faucet-Close | Faucet-Open | Handle-Press | Hammer | Assembly | Overall |
| Object Mask Proxy | 25.3% | 9.3% | 17.3% | 2.7% | 0.0% | 8.2% |
| AVDC (Full) | 53.3% | 24.0% | 81.3% | 8.0% | 6.7% | 43.1% |

Table 4: Object mask as Proxy for Action

Table 4 shows that our proposed AVDC outperforms the method that predicts the object masks as a proxy for the actions.

### D.4 QUALITY ANALYSIS ON SYNTHESIZED VIDEOS

We further quantitatively compared the synthesized videos to the ground truth videos regarding PSNR, SSIM, MSE, and LPIPS. Specifically, we synthesized videos with our trained Meta-World video model given unseen first frames (Unseen initial configurations), and compared every synthesized video frame to the corresponding ground truth video frame. We report average PSNR, SSIM, MSE, and LPIPS (AlexNet) scores over 15 videos for each view, totaling 15*3*11=495 videos being evaluated. Also, we report the scores comparing the last frames of synthesized videos and ground truth videos. The following table summarizes the result.

|  | PSNR ↑ | SSIM ↑ | MSE ↓ | LPIPS ↓ |
|---|---|---|---|---|
| Last Frame | 25.46 | 0.8920 | 0.0050 | 0.0525 |
| Whole Video | 25.02 | 0.8847 | 0.0057 | 0.0557 |

Table 5: Quantative Quality Analysis on Synthesized Videos

Table 5 shows that our proposed video diffusion model can reliably synthesize videos of task execution.

## E ADDITIONAL ABLATION STUDIES

### E.1 FIRST-FRAME CONDITIONING

To study the effectiveness of our first-frame conditioning strategy, we conducted a quantitative experiment that compares our RGB-channel-wise concatenating strategy with a trivial baseline: frame-wise concatenate, which concatenates the input frame before the first frame of the noisy video. We calculated the mean squared error (MSE) between the last frame of the ground video and the last frame of the synthesized video to evaluate the quality of synthesized videos. We show the results

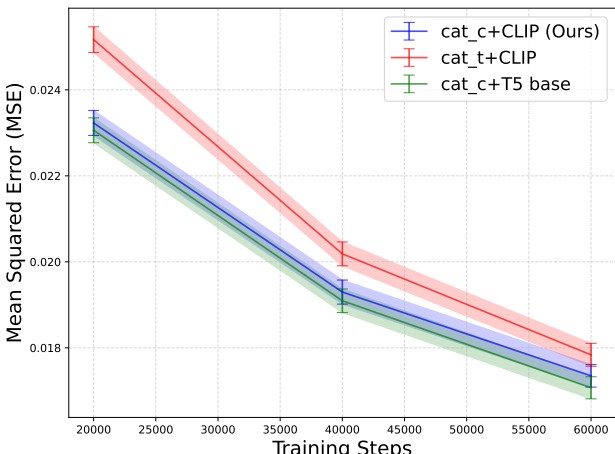

Figure 12: **Additional Ablation Studies on the First Frame Condition Strategy and Text Encoders.** We ablate the first frame conditioning strategy and compare the performance of different text encoders. Specifically, we calculate the MSE between the last frame of a ground truth video and the last frame of a synthesized video. **cat_c (Ours)**: the first frame is concatenated with a noisy video in RGB dimension, which is our proposed method. **cat_t**: the first frame is concatenated with a noisy video in time dimension. **CLIP**: CLIP text encoder (63M). **T5**: T5 base encoder (110M).

in Figure 12. Each data point is an average MSE calculated with 4000 samples of video generation, and the error bar shows the standard error of MSEs. Our method (cat_c) consistently outperforms frame-wise concatenating (cat_t) in the early stages of training on the Bridge dataset. Some qualitative generation results can be found on our supplementary website.

### E.2 TEXT ENCODER

We compare the video generation quality of our CLIP-text encoder (63M parameters) with the same model but with a T5-base encoder (110M parameters), dubbed AVDC (T5-Base). We used the same pixel-level MSE error as the evaluation metric. Figure 12 shows the result. The difference between the performance of the two text encoders is not significant.

### E.3 FACTORIZED SPATIAL-TEMPORAL CONVOLUTION

We compare the minimum VRAM requirement to train a video model in our setting with and without spatial-temporal convolution. The minimum requirement for training with a batch size of 1 in our Meta-World setting is 13139MB with factorized convolution and 16708MB without it. Our Bridge setting requires 8625MB with factorized convolution and 12606MB without it. In our Bridge experiments, we fine-tuned our video diffusion model on human data on a 3080-Ti GPU (12GB VRAM), which is only possible with factorized convolution.

## F   MODEL ARCHITECTURE AND TRAINING DETAIL

### F.1 TEXT ENCODER

We used a fixed pre-trained CLIP-Text encoder for encoding text descriptions. After encoding a text description with the encoder, we employ (Perceiver; Jaegle et al., 2021) as our attention-pooling network to aggregate the output tokens from the CLIP-text encoder into one single vector and added it to the time embedding of the diffusion model. This simple condition mechanism has been successful in our experiments. We did not use cross-attention on text inputs throughout this work. The hyperparameters of Perceiver are listed in Table 6.

### F.2 VIDEO DIFFUSION MODEL

For all models, we use dropout=0, num_head_channels=32, train/inference timesteps=100, training objective=predict_v, beta_schedule=cosine, loss_function=l2, min_snr_gamma=5, learning_rate=1e-4, ema_update_steps=10, ema_decay=0.999. We list all other hyperparameters in Table 7.

| Parameter | Value |
|---|---|
| **layers** | 2 |
| **num_attn_heads** | 8 |
| **num_head_channels** | 64 |
| **num_output_tokens** | 64 |
| **num_output_tokens_from_pooled** | 4 |
| **max_seq_len** | 512 |
| **ff_expansion_factor** | 4 |

Table 6: Model parameters for our Perceiver.

| | **Meta-World** | **iTHOR** | **Bridge** |
|---|---|---|---|
| **num_parameters** | 201M | 109M | 166M |
| **resolution** | (128, 128) | (64, 64) | (48, 64) |
| **base_channels** | 128 | 128 | 160 |
| **num_res_block** | 2 | 3 | 3 |
| **attention_resolutions** | (8, 16) | (4, 8) | (4, 8) |
| **channel_mult** | (1, 2, 3, 4, 5) | (1, 2, 4) | (1, 2, 4) |
| **batch_size** | 16 | 32 | 32 |
| **training_timesteps** | 60k | 80k | 180k |

Table 7: Comparison of configuration parameters for Meta-World, iTHOR, and Bridge.

# G  HARDWARE AND COMPLEXITY ANALYSIS

## G.1  TRAINING COST

We train all models on 4 V100 GPUs with 32GB memory each. For environment-specific datasets like those used in Meta-World and iTHOR experiments, the training of video policies can be finished within a day. As for a much larger dataset like Bridge, we have to train for longer to obtain consistent results. Despite the large size of Bridge data compared to the datasets we used in the other two experiments, we can generate high-quality and consistent results with just two days of training.

- Meta-World: about 24 hours of training (165 videos)
- iTHOR: about 24 hours of training (240 videos)
- Real-world experiment: about 48 hours of pre-training on around 40k Bridge videos and 4 hours of fine-tuning on 20 human videos.

## G.2  INFERENCE COST

We conducted our experiments (inference) on a machine with an RTX 3080Ti as GPU. We provide a detailed run time breakdown on Meta-World experiment of each step of our method below.

- **Text-Conditioned Video Generation**: Synthesizing a video of predicted execution is the most time-consuming step of our method, which takes roughly **10.57** seconds (**1.51** seconds per video frame on average).
- **Flow Prediction**: Predicting optical flow between a pair of two subsequent frames takes **0.28** seconds on average.
- **Action Regression from Flows and Depths**: Inferring the action from optical flow prediction **1.31** seconds on average.
- **Action Execution**: Running an inferred action using the controller in the environment takes **1.53** seconds on average.

## G.3  REPLANNING COST

In Meta-World experiments, AVDC used about 18 seconds for each round of action planning. Since the maximum number of replans is set to 5, the number of action planning rounds within an episode varies from 1 to 6. Therefore, the total planning cost ranges from about 18 to 108 seconds on a Nvidia GeForce RTX 3080Ti GPU.

### G.4 IMPROVING INFERENCE EFFICIENCY WITH DENOISING DIFFUSION IMPLICIT MODELS

We can incorporate various techniques into our method to improve its inference efficiency. To speed up the video synthesis step, we can progressively distill the diffusion models for faster sampling (Salimans & Ho, 2022). Also, we can leverage lighter-weight optical flow prediction models to increase efficiency. To accelerate action prediction from flows, we can design more sophisticated techniques for sampling and optimizing actions or parallelizing them using GPUs.

This section investigates the possibility of accelerating the sampling process using Denoising Diffusion Implicit Models (DDIM; Song et al., 2021). To this end, instead of iterative denoising 100 steps, as reported in the main paper, we have experimented with different numbers of denoising steps (*e.g.*, 25, 10, 5, 3) using DDIM. The qualitative results of synthesized videos are presented on our supplementary website.

We have found that reducing the number of denoising steps to 10 still leads to satisfactory generated video quality while resulting in a 10x speedup in video generation. Specifically, the overall mean success rate across tasks in Meta-World with 10-step DDIM is **37.5%**, which is competitive with our original method with 100 denoising steps with an overall success rate of **43.1%**. That said, when the task is running time-critical, we can speed up the video generation step by ten times with only **5.6%** drop in task performance.

## H DETAILS ON EXPERIMENTAL SETUP

This section describes experimental details, including learning diffusion models, inferring actions, replanning strategies, etc.

### H.1 META-WORLD EXPERIMENTAL SETUP

This section describes the details of the Meta-World experiments.

#### H.1.1 LEARNING THE DIFFUSION MODEL

We aim to learn a video diffusion model that can synthesize a video, showing a robot fulfilling a task, from an initial frame and a the task described in natural language. We found that in most goal-conditioned manipulation tasks, the final frame of the whole video is often highly correlated to the text description when the current (first) frame is given. In other words, the model can easily synthesize the last frame given the current frame and text description, while the model is often more uncertain about intermediate frames and therefore performs poorly in synthesizing intermediate frames.

To take advantage of this finding, we propose an *adaptable frame sampling technique* to sample frames from the whole video for training. In particular, we first randomly sample a frame from the whole video dataset as the current frame. We then uniformly sample $T - 2$ frames from the current (*i.e.*, initial) frame to the final frame from the same video. Then, we use these $T$ frames (1 current/initial frame, $T - 2$ intermediate frames, and 1 final frame) to train our video diffusion model. We empirically found that this *adaptable frame sampling technique* significantly improves the learning efficiency of the video diffusion model, enabling the training to finish within a single day using just 4 GPUs.

#### H.1.2 CALCULATING OBJECT RIGID TRANSFORMATIONS AND INFERRING ACTIONS

Given the predicted optical flow between each pair of frames of a synthesized video and the initial frame, we aim to infer a robot's actions to follow the synthesized video.

**Tracking Object and Determining Contact Point.** Since Meta-World focuses on manipulating objects, we propose to track an object of interest by extracting an object mask. To determine the contact point for the robot to grasp an object, we simply sample $N = 500$ points from the object mask and compute the centroid of an object as the contact point. Note that more sophisticated methods for determining contact points can be employed to further improve the proposed method.

**Calculating Object Rigid Transformation Object and Computing Subgoal.** Given the optical flow computed from the synthesized video frames, we can use it to compute the 2D correspondence

between two frames. We use the RANSAC algorithm to find an optimal 2D transformation that produces the most inliners from the 2D correspondences. We only use these inliner points for the computations in the current and the following steps for better robustness. We apply our method as described in Section 3.3 to obtain a sequence of 3D rigid transformations. Then, we apply these transformations on the sampled grasp sequentially to obtain a sequence of subgoals, indicating how the object should be moved.

**Inferring Actions.** Given each subgoal, we decide whether to use "grasp" action or "push" action to interact with the object by checking if the maximum magnitude of vertical displacement exceeds 10cm based on the heuristic that pick-and-place tasks usually require the robot to lift the object, which produces a vertical displacement; on the other hand, optimal object trajectories of pushing tasks do not exhibit such vertical displacement.

Once we decide if the robot should "grasp" or "push" the object, we determine the robot arm's action as follows. For the grasp mode, we simply control the robot to take a grasping action (closing the grippers) at the point and then move toward the subgoals. For the push mode, we put the robot arm in a specific direction to the object that allows pushing before moving the robot towards the subgoals. We calculate such direction by extrapolating the line between the push point and the first subgoal more than 10cm away from the grasp. Here, we consider the sampled grasp location as the contact point for the push action, referred to as the "push point".

### H.1.3 REPLANNING STRATEGY

In Meta-World, we replan (*i.e.*, perform the closed-loop control) by synthesizing a video with the current observed state as the initial frame for the video diffusion model. Then, we use the current object mask and depth information to compute a new sequence of subgoals. Note that we do not re-decide the interaction mode. For the grasp mode, we simply move the gripper toward the new subgoals. For the push mode, we re-initialize the gripper as described above. In specific, when re-planning is triggered, we re-initialize the gripper by 1) syntheiszing a video plan, 2) sampling grasp and calculating subgoals, and 3) calculating the direction for placing the gripper by extrapolating the line between the grasp point and the first subgoal. Once the re-initilization is done, we can start the robot execution. After the re-initialization, we start to move the gripper toward the new subgoals.

### H.1.4 DETAILS OF BASELINES

**Diffusion Policy** Following the original paper, the image observation is encoded with the adapted ResNet-18 with group norm and spatial softmax pooling. For the diffusion backbone model, we experimented with the 1D convolutional FiLM U-net architecture proposed in the paper. The backbone model is adapted to take in task embeddings, i.e., the CLIP-Text task embeddings are concatenated with the observation embeddings. The hyperparameters $T_o$ (the number of past frames used as a condition), $T_p$ (the number of future actions to predict), and $T_a$ (the number of actions to execute before replanning, $0 < T_a <= T_p$), were set to align with the configurations used in most of the paper's experiments. Specifically, we set $(T_o, T_p, T_a) = (2, 16, 8)$. We used a batch size of 4096 and evaluated the checkpoints at 15k, 25k and 35k training steps. We picked the checkpoint with the best overall performance as the Diffusion model baseline.

### H.2 ITHOR EXPERIMENTAL SETUP

This section describes the details of the iTHOR experiments.

### H.2.1 LEARNING THE DIFFUSION MODEL

We aim to learn a video diffusion model that can synthesize a video that shows an agent navigating to a target object in first-person point of view in iTHOR indoor scenes. To sample video segments for training, we first randomly sample a frame from the video demonstration dataset, and then we retrieve $T - 1$ consecutive future frames from the same video. We do not skip any intermediate frames (*i.e.*, we do not apply the *adaptable frame sampling technique* used in Meta-World), as iTHOR environment uses discrete actions such as moving and rotating for a constant distance or angle. If the number of subsequent frames is less than $T - 1$ in the sampled video, we duplicate the last frame to compensate for missing frames. This allows the model to recognize that the target object is found and the agent should stop moving.

### H.2.2 Calculating Scene Transformations and Inferring Actions

**Tracking Scene and Calculating Scene Transformation.** In the navigation setup, instead of tracking the correspondences of a particular object, we track the correspondences of the entire scene. To this end, instead of generating an object mask, we initialize a scene mask by thresholding out moving points (magnitude of optical flow $> 1$). Then, to calculate the scene transformation, we apply a similar procedure to the Meta-World experiment for computing the object transformation. However, we do not use the RANSAC algorithm to obtain inliners; instead, at each time step, we simply remove the key points that move out-of-bound (*i.e.*, outside the image) and keep the rest as the inliner points to calculate the scene transformation.

**Inferring Actions.** Given calculated the scene transformation at each step, we design a procedure to infer an action (`MoveForward`, `RotateLeft`, `RotateRight`, or `Done`). We propose to observe an *imaginary point* located 1 meter in front of the robot. We apply the calculated scene transformation on this *imaginary point*. We then decide the action based on the translation of this *imaginary point* before and after applying the transformation. Specifically, if the translation is close to 0 ($< 1$mm), since the agent stays still, we choose `Done`. Otherwise, we check the horizontal displacement of this *imaginary point*. If the magnitude of the horizontal displacement is less than 25cm, we choose `MoveForward`. Otherwise, we select `RotateLeft` or `RotateRight` depending on the direction of the displacement.

### H.2.3 Replanning Strategy

In iTHOR, we replan when we lose track of most correspondences from the initial frame. Specifically, if the number of inliners we keep is less than $10\%$ of the original number we sampled, we re-synthesize a video, predict optical flow, and infer actions.

### H.3 Real-World Franka Emika Panda Arm Experimental Setup

This section describes the details of the real-world Franka Emika Panda arm experiments.

**Hardware.** Our arrangement comprises a Franka Emika Panda arm and an Intel Realsense D435 RGBD camera mounted at a fixed frame relative to the table. The robot arm is equipped with a parallel motion two-jaw gripper, and the robot arm is in joint position control mode. Meanwhile, the camera has calibrated intrinsic and extrinsic. Therefore, any object motion predicted in the camera frame can be directly transformed into the world frame.

**Dataset Collection.** After training on the Bridge dataset, we fine-tuned the model with 20 human demonstrations in a real-world tabletop manipulation setting. These videos are collected by humans using their hands to move objects on the table and accomplish tasks. Our object set includes plates, bowls, a few categories of fruits (apples, oranges, bananas, peaches, and mangoes), and utensils such as forks and knives as distractors. The task is to pick up fruits from their initial locations and place them in the specified container, a plate or a bowl.

**Action Prediction and Execution.** In our real-world evaluation, we assume that the target object can be grasped using a top-grasp and that no re-orientation of the target object is needed. Therefore, in order to compute the target object poses, we first manually specify the segmentation of the object (in principle, it can be done using other object segmentation models, too), extract the corresponding optical flow, and compute the sequence of the object poses. We extract its initial pose (in the first frame) and the target pose (in the last frame), and generate a robot arm trajectory using an inverse-kinematics (IK) solver. In practice, we found that since we are using only a small set of tracking points (objects are small in our camera view), the reconstruction of 3D rotations is not robust. This can be potentially addressed by leveraging a higher-resolution video generative model or simply, different camera configurations.

**Failure Mode Analysis** In our real-world experiments, we found that our approach failed in 8 of the 10 tested trials. We found that 75% of the failures were caused by the wrong plan from the video diffusion model. It either picked the wrong object or placed it at the wrong target. The other 25% of the failures were caused by the discontinuity of video generation. The generated plan seems to be correct, but the object disappeared in some intermediate frame, which eventually led to the failure.

