# OpenReview forum: "Learning to Act from Actionless Videos through Dense Correspondences"
_ICLR.cc/2024/Conference — ICLR 2024 spotlight_

### Official Review · Reviewer_Hwk5 · 2023-10-31

**Soundness:** 4 excellent
**Presentation:** 4 excellent
**Contribution:** 3 good
**Rating:** 6
**Confidence:** 4

**Summary:**

The paper presents a method called Actions from Video Dense Correspondences (AVDC) that learns to perform robotic tasks without ever accessing labels of robot actions. The method first trains a text-conditioned diffusion video prediction model on videos of robot data. It then computes optical flows for adjacent frames of the prediction, and by segmenting out the object of interest and finding a rigid body transformation that best fits the corresponding points, computes an object (manipulation) or robot (navigation) transformation that enables the robot to follow the plan. Experiments are conducted on several environments: simulated meta-world and iTHOR tasks, visual pusher for evaluating learning from a human embodiment, and bridge data/a real Panda arm.

**Strengths:**

- Learning from action-free videos is an important and challenging problem, which could enable data-driven robotics to access scale through sources like Youtube.
- The proposed method makes sensible design decisions, including the video model and correspondence computation strategy.
- The evaluation is very thorough in terms of the number of environments considered, and AVDC appears to yield consistent performance gains.
- The authors have committed to making their video model, which requires significantly fewer resources than prior works, open source. I think this is also a valuable contribution (although less related to the main message of the work).
- The presentation is quite good, the writing is clear, and main ideas are communicated directly.

**Weaknesses:**

- The most apparent weakness of this work is that not all tasks that a robot might want to solve can be solved by a trajectory of target object poses. For example, it is unclear how to plan a task that would require the robot to use another object as a tool. It is also unclear how deformable objects could be handled. There also may be tasks such as pressing a button on a microwave, which do not involve the microwave moving, but require a particular amount of force to be applied, for which this may not be applicable.
- The work “Zero-Shot Robot Manipulation from Passive Human Videos” by Bharadhwaj et al. presents very similar (although not identical) ideas and is not discussed in the prior work or cited. Could the authors discuss and ideally perform a comparison to some of the ideas in that work?
- Related to the previous point, the ideas presented in this paper are not entirely novel. However, I believe that this particular combination/instantiation of them, as well as the evaluation and exploration of them that is provided, is a valuable contribution to the community.

**Questions:**

- The video generation performance is seemingly quite good even when very few training trajectories are provided (just 165 videos for Meta-World). Can you comment about how overfitting can be avoided or provide some intuition?
- The performance of the UniPi baseline is surprisingly poor. Could you please provide an explanation for the common failure modes or visualizations? Same goes for the BC baselines. Is this due to the low amount of data provided, thus causing action prediction models to overfit? If so, would it be possible to report results with greater number of demonstrations (like 50, or even 15, rather than 5)?
- Are the camera poses that are used for evaluating the policies the same as the ones in the training data? I assume that they are but it would be good to have confirmation.

---

> ### Author Response · Authors · 2023-11-18
> **Response to Reviewer Hwk5 (1/3)**
>
> We thank the reviewer for the thorough and constructive comments. Please find the response to your questions below.
>
> **Q1** Generalization to more complex tasks.
> > The most apparent weakness of this work is that not all tasks that a robot might want to solve can be solved by a trajectory of target object poses. For example, it is unclear how to plan a task that would require the robot to use another object as a tool. It is also unclear how deformable objects could be handled. There also may be tasks such as pressing a button on a microwave, which do not involve the microwave moving, but require a particular amount of force to be applied, for which this may not be applicable.
>
> **A:** We thank the reviewer for bringing up these points. We will revise the paper to discuss them.
>
> We would like to clarify that **tasks requiring the use of tools** can be indeed achieved with our proposed method. In particular, our method can achieve this by predicting the movements of a tool and rendering actions that induce such movements. For example, in Meta-World, the hammer task requires using a hammer as a tool.
>
> The current implementation of our method cannot complete tasks that require interacting with **deformable objects**. One possible extension would be to consider tracking key points (e.g., corners of a piece of cloth) and use the key points to recover robot motion. However, this requires additional knowledge, such as key points or other representations of deformable objects.
>
> Another limitation of learning-from-passive-video frameworks is the missing of **force information**. Therefore, for forceful manipulation tasks, additional training or models for forceful actions would be needed (but being able to predict that a button should be pushed down would still be useful).
>
> **Q2** Related work by Bharadhwaj et al.
> > The work "Zero-Shot Robot Manipulation from Passive Human Videos" by Bharadhwaj et al. presents very similar (although not identical) ideas and is not discussed in the prior work or cited. Could the authors discuss and ideally perform a comparison to some of the ideas in that work?
>
> **A:** We thank the reviewer for suggesting this paper. It is definitely sharing the same spirit and some ideas with our work. We will discuss this paper in the revision in detail. While this paper and our work tackle the problem of learning from passive videos, we notice three differences between our method and Bharadhwaj et al.
>
> First, our video prediction model is conditional on a textual description of the task. By contrast, Bharadhwaj et al. used an unconditional generation or goal-image conditioning.
> Second, we predict an object-centric trajectory given the initial frame; by contrast, Bharadhwaj et al. learned to predict hand poses. While hand poses contain more details of manipulator-object interactions, they may have cross-embodiment transfer issues. Finally, Bharadhwaj et al. directly predict pose trajectories, while ours predicts videos first. In our early experiments, we have found that predicting video frames gives better results than predicting object motion (via prediction optical flows). We posit that this is because video prediction is a dense prediction problem, and convolutional neural networks have been designed for this task; by contrast, optical flows are more sparse for most of the scenes because there are usually only a few objects moving in the scene.
>
> Unfortunately, comparing with Bharadhwaj et al. is not possible because the authors have not yet released their code/models at this moment.

---

> ### Author Response · Authors · 2023-11-18
> **Response to Reviewer Hwk5 (2/3)**
>
> **Q3** Explanation of why overfitting did not happen.
> > The video generation performance is seemingly quite good even when very few training trajectories are provided (just 165 videos for Meta-World). Can you comment about how overfitting can be avoided or provide some intuition?
>
> **A:** In general, we find that predicting trajectories with diffusion has less tendency to overfit than single forward models, as the diffusion model must learn the task of denoising multi-modal samples similar to trajectories of the training distribution (which gives much more supervision than directly learning a single state/action mapping). This can also be validated by comparing a simple BC baseline and Diffusion Policy. Here, we show an extended experiment below: a Diffusion Policy (Chi, Cheng, et al. "Diffusion policy: Visuomotor policy learning via action diffusion") with a similar network architecture outperforms the BC baseline.
>
> | method | Door Open | Door Close | Basketball | Shelf Place | Button Press | Button Press Topdown | Faucet Close | Faucet Open | Handle Press |  Hammer | Assembly | Overall |
> | -------------------------------------- | ---------:| ----------:| ----------:| -----------:| ------------:| --------------------:| ------------ | -----------:| ------------:| -------:| --------:| -------:|
> |           BC |        21.3 |         36.0  |            0.0 |           0.0   |           34.7 |                  12.0  |           18.7 |          17.3 |           37.3 |      0.0   |        1.3 |      16.2 |
> | Diffusion Policy                          |      45.3 |       45.3 |        8.0 |         0.0 |         40.0 |                 18.7 | 22.7|**58.7**|21.3|4.0|1.3|24.1|
> AVDC (Full) |**72.0**| **89.3** | **37.3** | **18.7** | **60.0** | **24.0** | **53.3** | 24.0 | **81.3** | **8.0** | **6.7** | **43.1** |
>
> However, the task performance of AVDC still surpasses the Diffusion Policy by a large margin. We posit that video prediction is more data-efficient way to learn behavior than behavior cloning-based methods because it only needs to learn to predict object motions (how things should move in a video). By contrast, BC-based methods, including Diffusion Policies, require "joint" learning of object motion and how to control robots.

---

> > ### Author Response · Authors · 2023-11-18
> > **Response to Reviewer Hwk5 (3/3)**
> >
> > **Q4** Baseline performance.
> > > The performance of the UniPi baseline is surprisingly poor. Could you please provide an explanation for the common failure modes or visualizations? Same goes for the BC baselines. Is this due to the low amount of data provided, thus causing action prediction models to overfit? If so, would it be possible to report results with greater number of demonstrations (like 50, or even 15, rather than 5)?
> >
> > **A:** We thank the reviewer for this suggestion. We included an additional experiment of training BC with more data. The results are reported below.
> >
> > |  [Method]-[# of demos] |   door-open |   door-close |   basketball |   shelf-place |   button-press |   button-press-topdown |   faucet-close |   faucet-open |   handle-press |   hammer |   assembly |   overall |
> > |------------|------------:|-------------:|-------------:|--------------:|---------------:|-----------------------:|---------------:|--------------:|---------------:|---------:|-----------:|----------:|
> > AVDC-165 |72.0| 89.3 | 37.3 | 18.7 | 60.0 | 24.0 | 53.3 | 24.0 | 81.3 | 8.0 | 6.7 | 43.1 |
> > |           BC-165 |        21.3 |         36.0  |            0.0 |           0.0   |           34.7 |                  12.0  |           18.7 |          17.3 |           37.3 |      0.0   |        1.3 |      16.2 |
> > |         BC-330|        21.3 |         65.3 |            0.0 |           0.0   |           45.3 |                   21.3 |           44.0 |          29.3 |           29.3 |      2.7 |        0.0   |      23.5 |
> > |         BC-660|        61.3 |         72.0 |            0.0 |           1.3 |           77.3 |                   49.3 |           77.3 |          77.3 |           62.7 |     10.7 |        0.0   |      44.5 |
> > |        BC-1650|        96.0 |         81.3 |            0.0 |           0.0   |           96.0 |                   85.3 |           93.3 |          94.7 |           86.7 |     12.0 |        0.0   |      58.7 |
> >
> > The experiment results above show that BC needs 20 videos per view (in total 660 videos with action labels, compared to AVDC trained with 165 videos without action labels) to perform similarly to our method AVDC (43.1% overall). That said, BC needs around four times more videos and action labels than our method, highlighting our proposed method's efficiency. Moreover, BC cannot solve most tasks that require grasping (e.g., pick and place, use tools) even with 50 demonstrations. We believe this indicates that the bottleneck of BC's performance is the number of demonstrations (or action labels). That said, BC's poor performance can be attributed to the low-data setting, causing the BC to overfit unwanted features and limiting it to generalizing to different initial configurations on various tasks.
> >
> > Similarly, while UniPi also predicts videos, it relies on an inverse dynamics model trained with action labels to recover actions. Therefore, it has the same limitations when training with a limited amount of data. We would like to emphasize that we included the performance of BC and UniPi to calibrate the difficulty of the tasks, and BC and UniPi have access to action labels that are not accessible to our proposed method; therefore, the comparison is not fair.
> >
> > **Q5** Camera poses in evaluation.
> > > Are the camera poses that are used for evaluating the policies the same as the ones in the training data? I assume that they are but it would be good to have confirmation.
> >
> > **A:** Yes. Three different camera poses in Meta-World are used for training and evaluation. We ran all the methods on all the three camera poses and reported the average performance. We will revise the paper to make it clear.

---

> > > ### Comment · Reviewer_Hwk5 · 2023-11-22
> > > **Response to Authors**
> > >
> > > Thank you for your thorough rebuttal and for addressing my questions. I continue to recommend acceptance.

---

### Official Review · Reviewer_itAQ · 2023-11-01

**Soundness:** 3 good
**Presentation:** 3 good
**Contribution:** 3 good
**Rating:** 8
**Confidence:** 4

**Summary:**

The paper introduces a method for constructing a video-based robot policy, capable of performing diverse tasks across different environments. This approach doesn't require action annotations but uses images for a task-agnostic representation. Text is employed for specifying robot goals. By synthesizing videos to predict robot actions and employing dense correspondences between frames, the model infers actions without explicit training labels. It can leverage the large-scale RGB videos on the internet for training, and use this knowledge for robotic manipulation. The paper showcases the effectiveness of this approach in tabletop manipulation and navigation tasks and also provides an open-source framework for efficient video modeling.

**Strengths:**

1. The paper proposed a new correspondence based method to obtain robot action in forecasted robot videos. It proves that a latent dynamic model is not needed if the forecasted video has good quality.
2. The authors proposed a new method to generate future videos using a diffusion model, which achieves efficient training. It provides a promising toolbox for the community.
3. The method is evaluated on two tasks, table-top manipulaion and in-door navigation, demonstrating its effectiveness in different domains.
4. The paper is well-written and solid.

**Weaknesses:**

1. The selected robot tasks are relatively toy, and the potential of such kind of video prediction method is not evaluated. However, this is not the weakness of this paper, but a common practice for video prediction based robot control.

**Questions:**

1. In the appendix H.1.2, the authors say "We calculate such direction by extrapolating the line between the grasp point and the
first subgoal more than 10cm away from the grasp" Should it be push point?
2. In Sec. H.1.3, "For the push mode, we re-initialize the gripper as described above " is not clear. What does the re-initialization refer to?
3. When learning the diffusion model for the IThor environment, why not apply the adaptable frame sampling technique  in this case?

---

> ### Author Response · Authors · 2023-11-18
> **Response to Reviewer itAQ**
>
> We thank the reviewer for the thorough and constructive comments. Please find the response to your questions below.
>
> **Q1** Grasp point => push point in H.1.2.
> > In the appendix H.1.2, the authors say "We calculate such direction by extrapolating the line between the grasp point and the first subgoal more than 10cm away from the grasp" Should it be push point?
>
> **A:** Yes. We thank the reviewer for catching this and will fix it in the revision.
>
> **Q2** Re-initialization in the push mode in H.1.3.
> > In Sec. H.1.3, "For the push mode, we re-initialize the gripper as described above "is not clear. What does the re-initialization refer to?
>
> **A:** When replanning is triggered, we re-initialize the gripper by (1) synthesizing a video plan, (2) sampling the grasp and calculating subgoals, and (3) calculating the direction for placing the gripper by extrapolating the line between the grasp point and the first subgoal. Once the re-initialization is done, we can start the robot execution. We will revise the paper to make this re-initialization procedure clear.
>
> **Q3** Why not applying adaptive frame sampling to iTHOR?
> >When learning the diffusion model for the IThor environment, why not apply the adaptable frame sampling technique in this case?
>
> **A:** We thank the reviewer for raising this question. We did not use adaptive sampling because iTHOR uses discrete actions such as moving and rotating for a constant distance or angle. We will clarify this in the paper.

---

### Official Review · Reviewer_cxfp · 2023-11-03

**Soundness:** 3 good
**Presentation:** 4 excellent
**Contribution:** 4 excellent
**Rating:** 10
**Confidence:** 3

**Summary:**

The paper presents an approach for learning video-based policies in robot manipulation settings. The key benefit of the approach is training on actionless video data across human and robot embodiments. The method, termed Actions from Video Dense Correspondences (AVDC), consists of three stages: (1) diffusion-based video prediction given a text-based goal and starting image, (2) optical flow prediction creating dense correspondences, and (3) executed on a robot platform using off-the-shelf inverse kinematics and motion planners. AVDC uses the ability to project a 3D point onto the image plane both from depth and optical flow to compute the transformation of rigid objects across the predicted video frames. These transformations allow AVDC to infer actions in the environment. Then off-the-shelf robotics primitives can be used to enact the planned trajectory. The approach is benchmarked on the Meta-World and iTHOR simulation platforms and on a real-world robot platform, outperforming the considered baselines.

**Strengths:**

* The general problem of making use of actionless human video data is of interest and importance to the research community.
* The problem is well-motivated and the literature review does a good job of contextualizing the paper in prior work.
* The paper is strong, well-written and easy to follow.
* The use of geometry to reconstruct the transformation of the predicted objects (stationary camera) or embodiment (moving camera) which can be derived simultaneously from the optical flow and depth camera during deployment is clever. This allows the training data for the video prediction and optical flow not to require depth, with depth only being necessary during deployment. The transformations of either the objects or the embodiment then can be used in conjunction with off-the-shelf inverse kinematics, motion planners, grasp point predictors, etc. This also allows for learning from human videos and then zero-shot deploying to the robot, which is very impressive.
* The figures are informative and effectively illustrate the benefits of the proposed approach.
* The experiments consider both simulation and real robot evaluation, as well as an ablation study, demonstrating AVDC's superior performance as compared to the considered baselines and support for AVDC's design choices. In particular, I appreciated the discussion and later the results for why not to directly predict the optical flow without the intermediate step of video prediction.
* The discussion did a good job of describing the weaknesses and failure modes of the proposed method.

**Weaknesses:**

* The literature review is missing a number of relevant works.
  * V-PTR: similar high-level motivation of using video-based, prediction-focused pre-training and then action-based finetuning. This should have likely served as a baseline for the proposed method.
  * [A] Bhateja, Chethan, et al. "Robotic Offline RL from Internet Videos via Value-Function Pre-Training." arXiv preprint arXiv:2309.13041 (2023).
  * Diffusion policy: diffusion policy has shown very good results in terms of multi-task, low-data regime performance.
    * [B] Chi, Cheng, et al. "Diffusion policy: Visuomotor policy learning via action diffusion." arXiv preprint arXiv:2303.04137 (2023).
    * [C] Ha, Huy, Pete Florence, and Shuran Song. "Scaling up and distilling down: Language-guided robot skill acquisition." arXiv preprint arXiv:2307.14535 (2023).
* In particular, my biggest concern with the paper is the lack of comparison to a strong BC baseline. As the AVDC method uses diffusion to predict images, it seems natural to baseline against a diffusion policy (e.g., [B, C]). R3M is a fairly old representation at this point (e.g., Voltron [D] would be a better representation) and the simple MLP-based BC policy would strongly underperform diffusion policy. This is confirmed by, for example, the very poor baseline performance in Tables 1 and 2. In Sec. 4.3, it is mentioned that since 'R3M is pretrained on robot manipulation tasks ... it might not be suitable for visual navigation tasks'. Something like [E] could be a better baseline here.
  * [D] Karamcheti, Siddharth, et al. "Language-driven representation learning for robotics." arXiv preprint arXiv:2302.12766 (2023).
  * [E] Shah, Dhruv, et al. "ViNT: A Foundation Model for Visual Navigation." arXiv preprint arXiv:2306.14846 (2023).
* In Sec. 4.5 'Results', the paper states that Fig. 10 presents screenshots of robot trajectories, but I believe that is Fig. 9? Fig. 10 shows human predicted trajectories.

Some typos and points of confusion are listed below:
1. Page 3 - 'Unipi'.
2. Sec. 4.1:  'compare AVDC to its [variants] that also predict dense correspondence'.
3. Sec. 4.2: 'maximum number of planning affects' -> 'maximum number of replanning steps affects'.

**Post-rebuttal: Most of my concerns have been addressed! I am raising my score as such.

**Questions:**

1. In the related work, you mention that RL based methods often have to interact with the environment. However, offline RL-based methods avoid this issue (e.g., [A]). What is the downside of such approaches compared to the proposed method?
2. Was the choice of the factorized spatial-temporal ResNet block ablated?
3. I did not quite understand in Sec. 3.3 'Predict object-centric motion', what happens to achieve subsequent subgoals after the first grasp-contact point is reached. Do you pick the next one in the subsequent predicted video frame?
4. In the replanning strategy, why would a smaller robot movement necessarily be indicative of failure? What if the inaccuracy in compounding error results in large, but inaccurate robot movements?
5. Is there a reason not to use a receding horizon-style replanning strategy as in [B]?
6. Do you have a sense as to why AVDC (Full) underperformed in the 'btn-press-top' task in Table 1?

---

> ### Author Response · Authors · 2023-11-18
> **Response to Reviewer cxfp (1/3)**
>
> We thank the reviewer for the thorough and constructive comments. Please find the response to your questions below.
>
> **Q1** Related work.
> > The literature review is missing a number of relevant works.
> > - V-PTR: similar high-level motivation of using video-based, prediction-focused pre-training and then action-based finetuning. This should have likely served as a baseline for the proposed method.
> >     - [A] Bhateja, Chethan, et al. "Robotic Offline RL from Internet Videos via Value-Function Pre-Training." arXiv preprint arXiv:2309.13041 (2023).
> > - Diffusion policy: diffusion policy has shown very good results in terms of multi-task, low-data regime performance.
> >     - [B] Chi, Cheng, et al. "Diffusion policy: Visuomotor policy learning via action diffusion."
> >     - [C] Ha, Huy, Pete Florence, and Shuran Song. "Scaling up and distilling down: Language-guided robot skill acquisition."
>
> > In particular, my biggest concern with the paper is the lack of comparison to a strong BC baseline. As the AVDC method uses diffusion to predict images, it seems natural to baseline against a diffusion policy (e.g., [B, C]). R3M is a fairly old representation at this point (e.g., Voltron [D] would be a better representation) and the simple MLP-based BC policy would strongly underperform diffusion policy. This is confirmed by, for example, the very poor baseline performance in Tables 1 and 2. In Sec. 4.3, it is mentioned that since 'R3M is pretrained on robot manipulation tasks ... it might not be suitable for visual navigation tasks'. Something like [E] could be a better baseline here.
> > - [D] Karamcheti, Siddharth, et al. "Language-driven representation learning for robotics." arXiv preprint arXiv:2302.12766 (2023).
> > - [E] Shah, Dhruv, et al. "ViNT: A Foundation Model for Visual Navigation." arXiv preprint arXiv:2306.14846 (2023).
>
> **A:** We appreciate the reviewer pointing out these relevant works. We will discuss them in the revised paper.
>
> **Video pre-training for robots (V-PTR)**: V-PTR learns from videos (the Ego4D dataset) without action labels by extracting a value function during Pre-Training Phase 1. Then, V-PTR learns a Q-value function during Pre-Training Phase 2, which requires actions. Finally, V-PTR is fine-tuned on downstream tasks, which requires learning by interacting with environments. In contrast, our proposed method requires no action labels or learning with environment interactions. With the differences in problem formulation, we believe V-PTR and our method are not comparable. Furthermore, the V-PTR paper was posted on arXiv on Sep 22, six days before the ICLR 2024 deadline (Sep 28), which makes it difficult for us to include a discussion or comparison to V-PTR by the time we submitted our work.
>
> **Diffusion Policy**: We thank the reviewer for suggesting this method. We have implemented and conducted experiments with the Diffusion Policy on Meta-World tasks. Following the paper, the image observation is encoded with the adapted ResNet-18 with group norm and spatial softmax pooling. For the diffusion backbone model, we experimented with the 1D convolutional FiLM U-net architecture proposed in the paper. The backbone model is adapted to take in task embeddings, i.e., the CLIP-Text task embeddings are concatenated with the observation embeddings. The hyperparameters $T_o$ (the number of past frames used as a condition), $T_p$ (the number of future actions to predict), and $T_a$ (the number of actions to execute before replanning, 0<$T_a$ <=$T_p$), were set to align with the configurations used in most of the paper's experiments. Specifically, we set $(T_o, T_p, T_a) = (2, 16, 8)$. The results are presented in the following table.
>
> | method | Door Open | Door Close | Basketball | Shelf Place | Button Press | Button Press Topdown | Faucet Close | Faucet Open | Handle Press |  Hammer | Assembly | Overall |
> | -------------------------------------- | ---------:| ----------:| ----------:| -----------:| ------------:| --------------------:| ------------ | -----------:| ------------:| -------:| --------:| -------:|
> |           BC |        21.3 |         36.0  |            0.0 |           0.0   |           34.7 |                  12.0  |           18.7 |          17.3 |           37.3 |      0.0   |        1.3 |      16.2 |
> | Diffusion Policy                          |      45.3 |       45.3 |        8.0 |         0.0 |         40.0 |                 18.7 | 22.7|**58.7**|21.3|4.0|1.3|24.1|
> AVDC (Full) |**72.0**| **89.3** | **37.3** | **18.7** | **60.0** | **24.0** | **53.3** | 24.0 | **81.3** | **8.0** | **6.7** | **43.1** |
>
> The results show that Diffusion Policy outperforms BC, and our method AVDC outperforms Diffusion Policy in 10/11 tasks and the overall performance by a large margin. Note that both BC and Diffusion Policy have access to action labels that are not accessible to our proposed method; therefore, the comparison is unfair. We will include the Diffusion Policy in our revised paper.

---

> > ### Author Response · Authors · 2023-11-18
> > **Response to Reviewer cxfp (2/3)**
> >
> > **Scaling up and distilling down** does not learn from videos; instead, it focuses on 6 DoF robotic skill learning, which differs from our problem formulation.
> >
> > **Language-driven representation learning for robotics** and **ViNT**: We thank the reviewer for suggesting these works that learn representations. To validate our BC implementation, we included an additional experiment of training BC with more data. The results are reported below.
> >
> > |  [Method]-[# of demos] |   door-open |   door-close |   basketball |   shelf-place |   button-press |   button-press-topdown |   faucet-close |   faucet-open |   handle-press |   hammer |   assembly |   overall |
> > |------------|------------:|-------------:|-------------:|--------------:|---------------:|-----------------------:|---------------:|--------------:|---------------:|---------:|-----------:|----------:|
> > AVDC-165 |72.0| 89.3 | 37.3 | 18.7 | 60.0 | 24.0 | 53.3 | 24.0 | 81.3 | 8.0 | 6.7 | 43.1 |
> > |           BC-165 |        21.3 |         36.0  |            0.0 |           0.0   |           34.7 |                  12.0  |           18.7 |          17.3 |           37.3 |      0.0   |        1.3 |      16.2 |
> > |         BC-330|        21.3 |         65.3 |            0.0 |           0.0   |           45.3 |                   21.3 |           44.0 |          29.3 |           29.3 |      2.7 |        0.0   |      23.5 |
> > |         BC-660|        61.3 |         72.0 |            0.0 |           1.3 |           77.3 |                   49.3 |           77.3 |          77.3 |           62.7 |     10.7 |        0.0   |      44.5 |
> > |        BC-1650|        96.0 |         81.3 |            0.0 |           0.0   |           96.0 |                   85.3 |           93.3 |          94.7 |           86.7 |     12.0 |        0.0   |      58.7 |
> >
> > The experiment results above show that plain BC needs 20 videos per view (in total 660 videos with action labels, compared to AVDC trained with 165 videos without action labesls) to perform similarly to our method AVDC (43.1% overall). That said, BC needs around 4 times more videos and action labels than our method, which highlights the efficiency of our proposed method. More importantly, BC still cannot learn most tasks that require grasps (e.g., pick and place, use tools) even with 50 demonstrations per view (BC-1650). We believe this indicates that the bottleneck of BC's performance is the number of demonstrations (or action labels), instead of learned representations.
> >
> > We would like to emphasize that we included the performance BC mainly for calibrating the difficulty of the tasks, and BC has access to action labels that are not accessible to our proposed method; therefore, the comparison is not fair.
> >
> > **Q2** Figure 10 clarifications.
> > > In Sec. 4.5 'Results', the paper states that Fig. 10 presents screenshots of robot trajectories, but I believe that is Fig. 9? Fig. 10 shows human predicted trajectories.
> >
> > **A:** We thank the reviewer for pointing this out. Figure 10 visualizes "robot planned trajectories" instead of "robot trajectories." We will revise the paper to fix this. The real-world robot execution trajectories can be found in section B or on our [supplementary website](https://flow-for-action-from-video.github.io/supplements/#realresults).
> >
> > **Q3** Typos.
> > > - Page 3 - 'Unipi'.
> > > - Sec. 4.1: 'compare AVDC to its [variants] that also predict dense correspondence'.
> > > - Sec. 4.2: 'maximum number of planning affects' -> 'maximum number of replanning steps affects'.
> >
> > **A:** We thank the reviewer and will fix the typos in the revision.
> >
> > **Q4** Offline RL does not interact with the environment.
> > > In the related work, you mention that RL based methods often have to interact with the environment. However, offline RL-based methods avoid this issue (e.g., [A]). What is the downside of such approaches compared to the proposed method?
> >
> > **A:** We thank the reviewer for raising this question. A critical difference between the problem formulations of offline RL methods and ours is that ours does not require any **action or reward labels**, while offline RL methods require both. We will revise the paper to make this clear.
> >
> > **Q5** Factorized spatial-temporal ResNet.
> > > Was the choice of the factorized spatial-temporal ResNet block ablated?
> >
> > **A:** The minimum VRAM requirement for training with a batch size of 1 in our Meta-World setting is 13139MB with factorized convolution and 16708MB without it. Our Bridge setting requires 8625MB with factorized convolution and 12606MB without it. In our Bridge experiments, we fine-tuned our video diffusion model on human data on a 3080-Ti GPU (12GB VRAM), which is only possible with factorized convolution. We will revise the paper to discuss this.

---

> > > ### Author Response · Authors · 2023-11-18
> > > **Response to Reviewer cxfp (3/3)**
> > >
> > > **Q6** Achieving subsequent subgoals after the grasp contact.
> > > > I did not quite understand in Sec. 3.3 'Predict object-centric motion', what happens to achieve subsequent subgoals after the first grasp-contact point is reached. Do you pick the next one in the subsequent predicted video frame?
> > >
> > > **A:** After reaching the first subgoal, we assume that the robot has grasped the object; therefore, we simply move the gripper to the next subgoal along the relative transformation calculated from subsequent video frames.
> > >
> > > **Q7** Would a small robot movement be indicative of failure?
> > > > In the replanning strategy, why would a smaller robot movement necessarily be indicative of failure? What if the inaccuracy in compounding error results in large, but inaccurate robot movements?
> > >
> > > **A:** In our design, a robot stops moving only when all the subgoals have been reached, or the robot gets stuck (e.g., when the subgoal is not reachable to the robot within a given timeout) and fails to reach the next subgoal. Therefore, the latter case triggers replanning by setting a small threshold to determine if a robot is stuck.
> > >
> > > **Q8** Receding horizon planning.
> > > > Is there a reason not to use a receding horizon-style replanning strategy as in [B]?
> > >
> > > **A**: One key advantage of the AVDC method's adaptive frame sampling is that AVDC video policy always plans to the end. This feature offers the potential to achieve the task objectives in a single video synthesis iteration, and in each retry. Given the substantial resources required for video synthesis, maximizing the task-related information represented by each video is essential. The adaptive frame sampling in AVDC gives global information about the task, rather than information only in the near future.
> > >
> > > Moreover, while video representation is more resource-intensive than action representation, it provides significantly greater temporal expressiveness. As shown in our experiments, AVDC can successfully solve various manipulation tasks, typically involving around a hundred environment steps (approximately 100 actions) by synthesizing just an 8-frame video.
> > >
> > > In summary, considering both the synthesis costs and the inherent capabilities of video representations, we believe the proposed AVDC style of replanning is a more efficient strategy for video policies.
> > >
> > > **Q9** AVDC on btn-press-top.
> > > > Do you have a sense as to why AVDC (Full) underperformed in the 'btn-press-top' task in Table 1?
> > >
> > > **A:** In this specific task, sometimes, a part of the button is located out of the image's border, which can cause the action prediction model to fail. In general, two potential issues might hurt the performance of our method more than the baselines:
> > > - Partial observability: It is difficult to generate the contact point of an object that is occluded by the robot or other objects.
> > > - Optical flow prediction failures: The quality of optical flow prediction could degrade when the moving object is at the edge of the image. AVDC (Flow) is less affected by this problem because we extracted the optical flow labels for training AVDC (Flow) with larger images (So that the button is never located out of the border) and then center-cropped the predicted flow to the size used in our setting.

---

> > > > ### Comment · Reviewer_cxfp · 2023-11-20
> > > > **Response to Rebuttal**
> > > >
> > > > Thank you for the thorough response to my review and the extra experiments! Some of these clarifications would be important to have in the updated paper draft and the extra experiments would make the paper stronger. Feel free to omit the Scaling up and distilling down - I agree it is not relevant here.
> > > >
> > > > Do you have any idea why BC achieves 0% on the basketball task?
> > > >
> > > > Assuming these updates to the paper, I will raise my score. Great work!

---

> > > > > ### Author Response · Authors · 2023-11-21
> > > > > **Re: Response to Rebuttal**
> > > > >
> > > > > We sincerely thank the reviewer for the timely response and increasing the score.
> > > > >
> > > > > > Do you have any idea why BC achieves 0% on the basketball task?
> > > > >
> > > > > The basketball task is challenging because the basketball is small and can easily slip due to inaccurate grasp or contact. We noted that even when 50 demonstrations were given, the performance of BC is still at 0% for many tasks that require grasping, including shelf-place, basketball, and assembly. We inspected the videos of BC policy execution and found that the primary failure mode is the policy's inaccurate grasp attempts. For example, the BC policy often closes the gripper early when it's still far from the basketball and cannot recover from this mistake, resulting in failing the task. In contrast, the Diffusion Policy can consistently make contact with the target object, and therefore, it succeeds in grasping more often than BC.
> > > > >
> > > > > We have updated our [supplementary website - Common Failure Modes of BC baselines](https://flow-for-action-from-video.github.io/supplements/#BCcommonfailuremode) to include the visualizations of the BC executions on shelf-place, basketball, and assembly. We hope this provides a better understanding of how BC fails.

---

### Official Review · Reviewer_QCWJ · 2023-11-10

**Soundness:** 3 good
**Presentation:** 3 good
**Contribution:** 3 good
**Rating:** 6
**Confidence:** 4

**Summary:**

The goal of this paper is to learn robot policies from action-free video data. The motivation is that there exists a lot of video data, but very little action data. Video prediction methods are often over-dependent on actions but have the benefit of being task agnostic. AVDC aims to solve this challenge by learning a video generation model (via diffusion) on the robot video data. From the generated sequence of images, the optical flow is estimated, which conditioned on some 3D knowledge as well as masks/segmentations of different objects gives an idea of the actions that are taken. The robot actions are then taken. To avoid accumulating errors, AVDC allows for replanning after a few actions. The approach is tested on manipulation (Meta-World) and navigation (iThor) setups, as well as qualitative results on a cross-embodiment visual pusher dataset and some robot arm data. Experiments and ablations find that (1) AVDC outperforms inverse dynamics+video prediction and BC baselines (2) all individual components are important.

**Strengths:**

- The paper tackles an important problem of learning from action-free videos
- The method, to my knowledge is novel
- The approach significantly outperforms baselines on many different tasks
- The ablations are well analyzed
- The paper is easy to follow and well written

**Weaknesses:**

- I think one of the main limitations is the setting: AVDC needs videos of robots performing the task. I believe this is a contrived setting as it is very likely that if video + 3D information is available, then this was a robot demonstration, and one can just collect action data. To me, it is unclear how this approach will scale beyond robot data.

- I am concerned by the reported results for the BC baseline. Due to action data being available, as well as the robot data being in-domain for the task a simple BC or kNN baseline should work very well. There are many cases where the results are < 5% success. This should be addressed. I would be willing to increase my score if this weakness is addressed.

- AVDC relies on object/robot masks - a simple baseline would be to use those as a proxy for the actions. One could get pseudo action labels from the videos and train a policy.

- AVDC assumes that all objects are going to be directly manipulated by the robot directly but this is not the when one object as a tool.

- Navigation approaches have many action free baselines which should be explored as well

- It would be good to see real world experiments

- It would be good to have more of an analysis on the quality of the video prediction model. I suspect it has a

**Questions:**

See weaknesses

---

> ### Author Response · Authors · 2023-11-18
> **Response to Reviewer QCWJ (1/2)**
>
> We thank the reviewer for the thorough and constructive comments. Please find the response to your questions below.
>
> **Q1** Setting limitations: robot video and 3D.
> > I think one of the main limitations is the setting: AVDC needs videos of robots performing the task. I believe this is a contrived setting as it is very likely that if video + 3D information is available, then this was a robot demonstration, and one can just collect action data. To me, it is unclear how this approach will scale beyond robot data.
>
> **A:** We would like to clarify the following two points:
> - **Our work does not need any 3D information during training**, and therefore, our proposed method learns from only videos with RGB frames. The 3D information is only required during inference for rendering robot actions.
> - **Our work can learn from videos of humans or robots that are different from the hardware setup performing actions.** Specifically, to show the ability of our proposed method to transfer across different embodiments, we have shown successful results with visual pusher experiments in Section 4.4, where the robot learns from *human video* (whose actions are not available). Additionally, in our real-world Franka Emica Panda arm experiment presented in Section 4.5, the robot learns from a large offline dataset from *other robots*, whose actions are not directly usable, and a small number of *human data* without action labels. AVDC only needs videos of an arbitrary agent performing tasks, such as a human or a different robot. This ability to learn from diverse data sources underscores its potential for real-world applications.
>
> **Q2** BC Baseline.
> > I am concerned by the reported results for the BC baseline. Due to action data being available, as well as the robot data being in-domain for the task a simple BC or kNN baseline should work very well. There are many cases where the results are < 5% success. This should be addressed. I would be willing to increase my score if this weakness is addressed.
>
> **A:** Thank you for the suggestion. To validate our BC implementation, we included an additional experiment of training BC with more data. The results are reported below.
>
> |  method-total_demos |   door-open |   door-close |   basketball |   shelf-place |   button-press |   button-press-topdown |   faucet-close |   faucet-open |   handle-press |   hammer |   assembly |   overall |
> |------------|------------:|-------------:|-------------:|--------------:|---------------:|-----------------------:|---------------:|--------------:|---------------:|---------:|-----------:|----------:|
> AVDC-165 |72.0| 89.3 | 37.3 | 18.7 | 60.0 | 24.0 | 53.3 | 24.0 | 81.3 | 8.0 | 6.7 | 43.1 |
> |           BC-165 |        21.3 |         36.0  |            0.0 |           0.0   |           34.7 |                  12.0  |           18.7 |          17.3 |           37.3 |      0.0   |        1.3 |      16.2 |
> |         BC-330|        21.3 |         65.3 |            0.0 |           0.0   |           45.3 |                   21.3 |           44.0 |          29.3 |           29.3 |      2.7 |        0.0   |      23.5 |
> |         BC-660|        61.3 |         72.0 |            0.0 |           1.3 |           77.3 |                   49.3 |           77.3 |          77.3 |           62.7 |     10.7 |        0.0   |      44.5 |
> |        BC-1650|        96.0 |         81.3 |            0.0 |           0.0   |           96.0 |                   85.3 |           93.3 |          94.7 |           86.7 |     12.0 |        0.0   |      58.7 |
>
> The experiment results above show that plain BC needs 20 videos per view (in total 660 videos with action labels, compared to AVDC trained with 165 videos without action labels) to perform similarly to our method AVDC (43.1% overall). That said, BC needs around four times more videos and action labels than our method, which highlights the efficiency of our proposed method. More importantly, BC still cannot learn most tasks that require grasping (e.g., pick and place, use tools) even with 50 demonstrations per view (BC-1650).
>
> We would like to emphasize that we included the performance of BC to calibrate the difficulty of the tasks, and BC has access to action labels that are not accessible to our proposed method; therefore, the comparison is not fair.

---

> ### Author Response · Authors · 2023-11-18
> **Response to Reviewer QCWJ (2/2)**
>
> **Q3** Baseline that uses pseudo action labels.
> > AVDC relies on object/robot masks - a simple baseline would be to use those as a proxy for the actions. One could get pseudo action labels from the videos and train a policy.
>
> **A:** We thank the reviewer for the suggestion. However, this is not directly applicable to our setting because computing robot actions requires depth, and we do not assume depth information during training, and none of the Bridge and Visual Pusher datasets have depth data.
>
> Still, to provide an idea of the performance of using object masks as a proxy for actions as suggested by the reviewer. We have conducted experiments with the following setting: We trained a model that takes in a segmented object and directly predicts optical flow within the segmentation (without diffusion). Then, we used the procedure as AVDC to calculate actions. The results are presented as follows.
>
> ||   door-open |   door-close |   basketball |   shelf-place |   button-press |   button-press-topdown |   faucet-close |   faucet-open |   handle-press |   hammer |   assembly |   overall |
> |---|------------:|-------------:|-------------:|--------------:|---------------:|-----------------------:|---------------:|--------------:|---------------:|---------:|-----------:|----------:|
> |Object mask proxy|1.3|20.0|0.0|0.0|12.0|2.7|25.3|9.3|17.3|2.7|0.0|8.2|
> |AVDC (Full) |72.0| 89.3 | 37.3 | 18.7 | 60.0 | 24.0 | 53.3 | 24.0 | 81.3 | 8.0 | 6.7 | 43.1 |
>
> The results show that our proposed AVDC outperforms the method that predicts the object masks as a proxy for the actions. We will revise the paper to include this experiment.
>
> **Q4** Not applicable for tasks that require using tools.
> > AVDC assumes that all objects are going to be directly manipulated by the robot directly but this is not the when one object as a tool.
>
> **A:** We would like to clarify that tasks requiring using tools can be indeed achieved with our proposed method. In particular, our method can achieve this by predicting the movements of a tool and rendering actions that induce such movements. For example, in Meta-World, the hammer task requires using a hammer as a tool.
>
> **Q5** Action-free baseline for navigation.
> > Navigation approaches have many action free baselines which should be explored as well
>
> **A:** We thank the reviewer for the suggestion. However, we are unsure what the "action-free" navigation baselines mean. Can the reviewer kindly provide specific papers or instruct on conducting additional comparisons?
>
> **Q6** Real-world experiments.
> > It would be good to see real world experiments
>
> **A:** We have shown real-world experiments with a Franka Emika Panda arm in Section 4.5; a further discussion can be found in Appendix H.3. Please also kindly see our videos in the [supplementary website](https://flow-for-action-from-video.github.io/supplements/#realresults).
>
> **Q7** Video prediction quality.
> > It would be good to have more of an analysis on the quality of the video prediction model. I suspect it has a
>
> **A:** We thank the reviewer for the suggestion. To measure the quality of synthesized videos, we provide qualitative results in Figure 5, Figure 7, Figure 8, Figure 10, and Appendix G.4. Also, we quantify the quality of synthesized videos by reporting the MSE between the last frames of synthesized videos and ground truth videos, presented in Appendix E and on the [supplementary website](https://flow-for-action-from-video.github.io/supplements/#vidgencomparison). We believe this sufficiently justifies the quality of the synthesized videos.
>
> Since the reviewer asked for additional analyses, we further quantitatively compared the synthesized videos to the ground truth videos regarding PSNR, SSIM, MSE, and LPIPS. Specifically, we synthesized videos with our trained Meta-World video model given unseen first frames, and compared every synthesized video frame to the corresponding ground truth video frame. We report average PSNR, SSIM, MSE, and LPIPS (AlexNet) scores over 15 videos for each view, totaling 15\*3\*11=495 videos being evaluated. Also, we report the scores comparing the last frames of synthesized videos and ground truth videos. The following table summarizes the result.
>
> ||PSNR ↑|SSIM ↑|MSE ↓|LPIPS ↓|
> |---|---:|---:|---:|---:|
> |last frame|25.46|0.8920|0.0050|0.0525|
> |whole video|25.02|0.8847|0.0057|0.0557|
>
> The results show that our proposed video diffusion model can reliably synthesize videos of task execution.

---

> ### Author Response · Authors · 2023-11-22
> **Looking forward to your feedback and discussion**
>
> Dear Reviewer QCWJ
>
> Thank you for reviewing our submission and your valuable feedback. We hope our clarifications on problem settings and new results (BC with more data, pseudo-action labels, and qualitative and quantitative results for video generation) can address your concerns. We are happy to discuss with you and answer any further questions. As the deadline for discussion is approaching, we very much look forward to your feedback.
>
> Thanks,
> Authors

---

> > ### Comment · Reviewer_QCWJ · 2023-11-22
> > **Response to rebuttal**
> >
> > Dear authors,
> >
> > Thanks for the thorough rebuttal. I have increased my score to 6.
> >
> > Best regards,
> > Review QCWJ

---

### Author Response · Authors · 2023-11-22
**Paper Revision**

We thank all the reviewers for their thorough and constructive comments. We have revised the paper to include these discussions. The major changes are summarized as follows.

1. **[Related works]** We thank the reviewers for pointing out various related works. We have revised the paper to discuss them.
    - Offline-RL: We clarified our problem formulation differs from the offline RL setup. (Reviewer cxfp)
    - "V-PTR": We discussed this recent approach that also learns from unlabeled video data. (Reviewer cxfp)
    - "Zero-Shot Robot Manipulation from Passive Human Videos": We discussed this approach that shares a similar spirit with our work. (Reviewer Hwk5)
    - Learning representations for robot learning: We included a discussion on more recent approaches that learn representations for robotics. (Reviewer cxfp)

2. **[Diffusion Policy]** (Reviewer cxfp and Reviewer Hwk5) As suggested by Reviewer cxfp, we have included comparisons to the Diffusion Policy in the Meta-World experiments, which justifies the effectiveness of our proposed method.

3. **[BC with more data]** (Reviewer QCWJ and Reviewer Hwk5) To calibrate the difficulty of the tasks, we have included additional experiments in Section D.2, which train BC with more demonstrations of the Meta-World tasks.

4. **[Object mask as a proxy for actions]** (Reviewer QCWJ) We have included an experiment using object masks extensively as a proxy for actions in Section D.3.

5. **[Additional analysis on synthesized videos]** (Reviewer QCWJ) We included a quantitative assessment of the synthesized videos in our Meta-World setting in Section D.4.

6. **[Tool-using tasks]** (Reviewer QCWJ and Reviewer Hwk5) In Section 4.2, we clarified the applicability of our method on tasks that require using tools.

7. **[Deformable objects]** (Reviewer Hwk5) We discussed the limitation of our approach to manipulate deformable objects and possible extensions to overcome the problem in the **Limitations** Section.

8. **[Clarification]**
    - Clarified that the same set of camera poses is used in training and testing in the Meta-World experiments. (Reviewer Hwk5)
    - Replaced "screenshots of robot trajectories" with "visualizations of planned robot trajectories." (Reviewer cxfp)
    - Explained why we did not use adaptable frame sampling as used in Meta-World experiments in Section H.2.1. (Reviewer itAQ)
    - Clarified the re-initialization procedure in Section H.1.3. (Reviewer itAQ)
    - Clarified the "push point" in Section H.1.2. (Reviewer itAQ)

---

### Comment · Area_Chair_A1t1 · 2023-11-23
**Author-Reviewer discussion period ending *very* soon**

Thank you very much to reviewers for responding. The authors have put great effort into their response, so can I please urge reviewer itAQ to answer the rebuttal.
Thank you!

---

### Meta-Review · Area_Chair_A1t1 · 2023-12-05

**Metareview:**

The paper introduces a video-based robot policy creation method that learns from minimal video demonstrations without explicit action annotations. It utilises images as a task-agnostic representation, capturing both state and action information, along with text for specifying robot goals. Through synthesising videos that simulate robot actions and leveraging dense correspondences between frames, the approach infers actions for execution without explicit action labels. This unique capability enables training policies solely from RGB videos, applicable across diverse robotic tasks. There is consensus among reviewers that this paper should be accepted (6, 6, 8, 10).

**Justification For Why Not Higher Score:**

Despite positive reviews, 2 reviewers did not raise their score above a marginal accept. Reviewer Hwk5's concern on novelty did not seem to get directly addressed, and reviewer QCWJ did not comment if their concern regarding applicability of the approach would scale beyond robot data.

**Justification For Why Not Lower Score:**

N/A

---

### Decision · Program_Chairs · 2024-01-16

Accept (spotlight)